# Rockfish: A transformer-based model for accurate 5-methylcytosine prediction from nanopore sequencing

Dominik Stanojević [1,2], Zhe Li [1], Sara Bakić[1,3], Roger Foo [4] & Mile Šikić [1,2] ✉

DNA methylation plays an important role in various biological processes, including cell differentiation, ageing, and cancer development. The most important methylation in mammals is 5-methylcytosine mostly occurring in the context of CpG dinucleotides. Sequencing methods such as whole-genome bisulfite sequencing successfully detect 5-methylcytosine DNA modifications. However, they suffer from the serious drawbacks of short read lengths and might introduce an amplification bias. Here we present Rockfish, a deep learning algorithm that significantly improves read-level 5-methylcytosine detection by using Nanopore sequencing. Rockfish is compared with other methods based on Nanopore sequencing on R9.4.1 and R10.4.1 datasets. There is an increase in the single-base accuracy and the F1 measure of up to 5 percentage points on R.9.4.1 datasets, and up to 0.82 percentage points on R10.4.1 datasets. Moreover, Rockfish shows a high correlation with whole-genome bisulfite sequencing, requires lower read depth, and achieves higher confidence in biologically important regions such as CpG-rich promoters while being computationally efficient. Its superior performance in human and mouse samples highlights its versatility for studying 5-methylcytosine methylation across varied organisms and diseases. Finally, its adaptable architecture ensures compatibility with new versions of pores and chemistry as well as modification types.

5-Methylcytosine (5mC) is among the most abundant and biologically relevant modifications involved in epigenetic regulation. In mammalian cells, DNA methylation contributes to maintaining genomic stability[1] and cellular functions[2], such as X-chromosome inactivation[3], transposon silencing[4], and genomic imprinting[5]. The landscape of DNA methylation is dynamically reprogrammed during development[6], and aberrant methylation patterns have been linked with diseases[7]. DNA methylation can also be influenced by various factors, including demographics (age, gender, race, etc.), environmental exposures (such as persistent organic/air/ heavy metal pollutants), and other risk factors (e.g. lifestyle and dietary exposures)[8].

On average, the frequency of 5mC in mammalian cells is 2-5% of all cytosine sites[9]. In the human genome, 5mC is mostly present at CpG dinucleotides (where p stands for phosphodiester bond) outside CpG islands which are regions of at least 200 bp, with at least 50% GC content and 60% observed-to-expected CpG ratio[10].

The gold standard method for 5mC detection at single-base resolution is bisulfite treatment[11]. Bisulfite treatment quickly converts cytosine to uracil, whereas 5mC is not influenced which leads to differential readouts in the sequencing[12]. A combination of the bisulfite treatment and NGS, also called whole genome bisulfite sequencing (WGBS) is a popular sequencing method. However, NGS sequencing

[1]Genome Institute of Singapore, Agency for Science, Technology and Research (A*STAR), Singapore, Singapore. [2]Faculty of Electrical Engineering and Computing, University of Zagreb, Zagreb, Croatia. [3]School of Computing, National University of Singapore, Singapore, Singapore. [4]Yong Loo Lin School of Medicine, National University of Singapore, Singapore, Singapore. ✉e-mail: mile_sikic@gis.a-star.edu.sg

produces reads only hundreds of base pairs (bp) long resulting in an unreliable alignment in repetitive genome regions, which constitute over 66% of the human genome[13]. Bisulfite conversion of unmethylated cytosines to uracils reduces sequence complexity, influencing sequence alignment. Moreover, the strand scission and formation of abasic sites induced by the harsh bisulfite reaction condition renders up to 99.9% of DNA fragments unsequenceable[14], and therefore sample information may get lost during sequencing. The DNA strands need to be amplified after bisulfite conversion to get enough material for sequencing and may introduce amplification bias.

Long-read technologies such as those of Oxford Nanopore Sequencing (ONT) and Pacific Biosciences (PacBio) enable direct sequencing of modified nucleotides which eliminates drawbacks of labor-intensive bisulfite treatment.

Methods using PacBio sequencing rely on the differences in the interpulse duration and pulse width between canonical and modified bases[15]. However, these methods are less accurate compared with the methods using ONT raw signal[16,17]. Moreover, CSS reads generated by PacBio sequencing are limited in length (mean length of 15 kbp or 24 kbp) and unable to bridge long repetitive regions[18]. In contrast, the ultra-long protocol in nanopore generates reads with N50 over 100 kbp[19], significantly longer than Illumina or CCS reads.

Nanopore sequencing[20] ratchets a DNA strand through the pore and measures the disruption of electrical current caused by the traversing DNA through the pore. The raw signals obtained from nanopore sequencing have been used to detect modifications by inspecting differences in ionic current between modified and unmodified bases[21,22]. Various approaches to detecting 5mC CpG modification using raw nanopore signals can be grouped into 3 categories: statistical testing, hidden Markov models and deep learning.

Nanoraw[23] and NanoMod[24] are tools that use statistical testing to detect modifications by comparison of unmodified and modified samples. Nanoraw uses the Mann–Whitney U-test combined with Fisher's method to group neighboring p-values. NanoMod replaces the Mann–Whitney U-test and Fisher's method with the Kolmogorov–Smirnov test and Stouffer's method, respectively. NanoRaw has been deprecated in favor of the ONT Tombo suite, which does not require an unmodified sample to detect modifications.

Nanopolish[21] and SignalAlign[25] are based on Hidden Markov Models (HMM) for detecting 5mC modifications. Nanopolish compares the likelihoods of both unmodified and modified k-mers which contain at least one CpG. If multiple CpGs are present in a k-mer, only k-mer level prediction is performed. SignalAlign uses HMM with a hierarchical Dirichlet process to learn modification effects in the raw current signal.

DeepSignal[26], DeepMod[27], Guppy, or Dorado[28], coupled with Remora[29], Rerio[30], and methBERT[31], are deep learning-based models for modification detection. While DeepMod utilizes (long short-term memory) LSTM architecture, DeepSignal combines LSTM and CNN (convolutional neural networks) architecture. DeepSignal2[32] replaces CNN with another LSTM, thereby reducing computational time. Guppy is a basecaller based on recurrent neural networks (RNN) architecture that adds a modified base to the canonical alphabet and performs sequence basecalling. Recently, ONT developed a new tool named Remora to decouple modification calling from canonical basecalling. After canonical basecalling, Remora performs a second, lightweight pass through the sequence and calls modifications. An alternative to the decoupled approach is Rerio, a research, Guppy-compatible basecaller that performs canonical and modified basecalling simultaneously. Megalodon[33] is another ONT-based tool built on top of Guppy (and Remora) callers. Megalodon anchors basecalling information to the reference sequence and uses called probabilities obtained from Guppy (and Remora) or Rerio alongside the basecalled and reference sequence to further increase the detection performance. In the rest of the manuscript, we use the term "Megalodon Remora" to refer to the pipeline which includes Guppy, Remora and Megalodon, "Megalodon Rerio" to denote the pipeline which consists of Rerio and Megalodon, and "Remora" to denote the pipeline which consists of Dorado and Remora. Both the default methylation Guppy model and the Rerio research model jointly call canonical and modified bases. Since Megalodon Rerio pipeline has been proven to outperform Guppy on R9.4.1 datasets[34,35], we evaluate Rerio research model in combination with Megalodon. For R10.4.1 datasets, we compare Rockfish against Dorado, as Guppy has been deprecated in favor of it. methBERT is a tool based on BERT[36], a large language model (LLM) based on the self-attention mechanism that achieved a significant breakthrough in many natural language processing (NLP) tasks and served as the baseline architecture in many other fields.

Most described methods based on nanopore sequencing require a higher coverage to correctly predict site-level methylation frequency due to their lower read-level prediction accuracy. In addition, precise read-level detection is essential when particular sites in the sample are not completely methylated or unmethylated, including differences between haploids and different cell types. Additionally, most previously described methods engage in heavy preprocessing where statistical descriptors (i.e., mean, standard deviation, length of signal) are extracted and used as input features rather than (or alongside) raw signal data. Furthermore, some of the aforementioned methods also heavily rely on recurrent neural architectures that have a bias towards the start and end tokens which influence the output the most[37]. Given that the usual preprocessing steps in previous methylation detection methods follow the setup of centering the sample sequence around the target position rather than having it at the beginning or the end of the sequence, recurrent neural networks might not be the best architectural choice for this problem. Furthermore, methBERT does not fully leverage the capabilities of the BERT architecture, as it primarily relies on signal statistics such as mean, standard deviation, and event length without incorporating the raw signal data. Additionally, methBERT, along with DeepSignal and DeepMod, depends on an event table generated by Tombo's re-squiggling algorithm. However, this algorithm is deprecated and no longer under active development, which poses a challenge for these methods. Recently, an upgrade of the DeepMod framework, named DeepMod2[38], was published. DeepMod2 extends the DeepMod framework by adding a Transformer-based model alongside the LSTM model. Similarly to methBERT, the Transformer-based DeepMod2 model is based on BERT-like architecture and relies on statistical descriptors of the nanopore signal. Due to the very recent publication of the DeepMod2 method, it was not included in our analysis. Notably, our R9.4.1 Rockfish model was evaluated in the DeepMod2 study and performed similarly or better than competitive tools in various contexts. Moreover, the DeepMod2 study confirms that Rockfish clearly outperforms other tools on the mouse dataset.

Considering the need for a highly accurate method for read-level prediction, we set out to develop a new, state-of-the-art deep learning method using modern architecture, Transformers. Our method, Rockfish, relies on raw nanopore signal, nucleobase sequence and alignment information to detect 5mC modification. We trained our model using high-quality human and mouse datasets and tested it on several R9.4.1 and R10.4.1 datasets including internally sequenced R9.4.1 H1 embryonic stem cell (H1ESc) native dataset and both R9.4.1 and R10.4.1 neonatal mouse (C57BL/6 Neonatal) data, and a few publicly available human cancer and blood datasets. Given that both R9.4.1 and R10.4.1 NA12878 and neonatal mouse datasets were used for evaluation, we indicate the pore version to differentiate between them. The remaining datasets were only sequenced with the R9.4.1 pore version. Rockfish models were extensively evaluated and compared with Megalodon Remora, Megalodon Rerio and Nanopolish for R9.4.1 datasets, and Remora for R10.4.1 datasets in the following six aspects: read-level prediction, site-level prediction, site-level correlation with

WGBS, calling coverage, execution time and resource utilization. In their assessment, Liu et al.[34] recommended Nanopolish for methylation analysis in the case of limited resources and Megalodon in the case of access to high-performance computing resources, therefore, those tools were chosen for comparison on R9.4.1 datasets. Similarly, since Remora is considered state-of-the-art for detecting 5mC on R10.4.1 data, Rockfish was compared with Remora on R10.4.1 datasets.

To summarize, our contributions in this paper are as follows:

- We introduce Rockfish, a Transformer-based model that achieves state-of-the-art accuracy in 5mC methylation detection in CpG context.
- We release several datasets that can be utilized for training or evaluating methylation detection tools.
- We perform an extensive evaluation against top-performing methylation detection tools.
- We release pre-trained base and small Rockfish models for R9.4.1 with 52.9M and 4.4M parameters, respectively, and a base Rockfish model for R10.4.1 with 34.7M parameters.

## Results

### Rockfish significantly improves read-level methylation prediction

Rockfish (Fig. 1) predicts read-level 5mC probability for CpG sites. The model consists of signal projection and sequence embedding layers, a deep learning Transformer model used to obtain contextualized signal and base representation and a modification prediction head used for classification. Attention layers in Transformer learn optimal contextualized representation by directly attending to every element in the signal and nucleobase sequence. Moreover, the attention mechanism corrects any basecalling and alignment errors by learning optimal signal-to-sequence alignment. During training, we introduced auxiliary tasks such as base prediction and signal classification tasks to further improve performance and generalization. For R9.4.1, we first trained a base model (teacher), and then further trained a "small model" (student) using knowledge distillation to reduce the running time and improve generalization. Both R9.4.1 models are collectively referred to as "Rockfish models" in this paper. Furthermore, we trained an R10.4.1 base model.

Nanopore sequencing predicts methylation at a single-base, single-strand resolution. First, we evaluated Rockfish against Megalodon coupled with Rerio, Megalodon coupled with Remora and Nanopolish on six R9.4.1 datasets, and against Remora on two R10.4.1 datasets on a read-level prediction. The results for six datasets are presented in Fig. 2a while the results for the remaining two datasets, NA19240 and H1ESc, are available in Supplementary Data 1. To ensure fair evaluation, we used only the read-level examples called by all ONT-based tools at sites covered by WGBS as the ground truth. Only fully unmethylated and fully methylated positions were used for evaluation. ONT-based tools are evaluated both genome-wide and in different genomic contexts (details in Methods-evaluation section): (1) singletons and non-singletons, (2) genic regions, (3) repetitive regions, (4) CpG islands, shores and shelves and (5) different GC content levels.

Read-level genome-wide results for all datasets are summarized in Fig. 2a. Both the base model and the small model significantly outperform Nanopolish and Megalodon. The R9.4.1 results show that the base model achieves 2.96 percentage points (further referred to as "pp") higher mean accuracy versus the second-best tool (Megalodon Remora for H1ESc and HX1, Megalodon Rerio for rest), and the small model increases mean accuracy by 3.09 pp. In the R10.4.1 case, the Rockfish model achieves 0.74 pp higher mean accuracy than Remora. Moreover, it can be seen that R9.4.1 Rockfish models generalize well on unseen datasets corresponding to the same cell type (NA12878, NA19240, and HX1), on unseen datasets corresponding to the different

cell types but the same species (H1ESc, K562), and even on unseen datasets corresponding to a different species (Mouse).

Besides genome-wide evaluation, we performed a read-level evaluation for various biological contexts Fig. 2b, c, full results in Supplementary Data 1). We evaluated Rockfish for singleton (CpG sites with only one CpG up and down 10-base-pair regions), and non-singleton (CpG sites with multiple CpG sites up and down 10-base-pair regions) examples. In both scenarios, R9.4.1 Rockfish models outperform Nanopolish and Megalodon models while R10.4.1 Rockfish model outperforms Remora. Almost all ONT-based tools achieve higher accuracy for non-singleton examples since models capture information from multiple CpG sites, making prediction easier. The only exception is Remora in the R10.4.1 case which achieves slightly better results in singletons compared to non-singletons.

Figure 2b shows the accuracy of different ONT-based tools on the R9.4.1 NA12878 dataset in different genic regions: promoters, exons, introns and intergenic regions. Both the base model and the small model outperform other ONT-based tools in these genic contexts. The weighted average error rate reductions across all R9.4.1 datasets of Rockfish base compared with the second-best tool range from 3.45 pp for promoters to 3.96 pp for exons. Rockfish small increases some of these differences even more - reduction values range from 3.43 pp for promoters to 4.0 pp for exons. The error rate reduction of Rockfish base compared with Remora in genic regions of R10.4.1 NA12878 varies from 0.52 pp in exons to 1.16 pp in promoters. Figure 2c shows results for different types of repetitive regions: long interspersed nuclear elements (LINEs), short interspersed nuclear elements (SINEs), long terminal repeats (LTRs), DNA transposons and "Other" (includes all types of repetitive regions not listed before, e.g. RNA Repeats). Both R9.4.1 Rockfish models achieve higher accuracy than Nanopolish and Megalodon. Similar to genic context analysis, both Rockfish models notably reduce the error rate compared with the second best tool within LINEs, SINEs, LTRs, and DNA transposons. In other repetitive regions, Rockfish models achieve a similar weighted average error rate to Megalodon Rerio, outperforming Megalodon Remora and Nanopolish. Furthermore, R10.4.1 Rockfish base outperforms Remora in all repetitive regions on the R10.4.1 NA12878 dataset.

Next, we evaluated Rockfish performance at CpG islands, shores, and shelves. Rockfish models significantly outperform Nanopolish and Megalodon on all R9.4.1 datasets (Supplementary Data 1). R9.4.1 Rockfish models achieve especially high accuracy at CpG islands (>98.17% with Rockfish base for all datasets, and >98.07% with Rockfish small for all datasets). Similarly, Rockfish outperforms Remora in CpG islands, shelves, and shores on the R10.4.1 NA12878 dataset with the highest difference of 2 pp in CpG islands. Lastly, we evaluated Rockfish at five different GC contents: 20%, 40%, 60%, 80% and 100%. Both R9.4.1 Rockfish models outperform Nanopolish and Megalodon for all GC contents. Similarly, R10.4.1 Rockfish outperforms Remora on R10.4.1 NA12878 dataset in all GC contents (Supplementary Data 1). All Rockfish models achieve higher accuracy for higher GC contents.

We then moved on to compare models' precision and recall (PR) for different probability thresholds. Figure 2d, e show precision-recall curves for R9.4.1 NA12878 and Mouse datasets that correspond to two different species and contain balanced and imbalanced read-level methylation distribution. R9.4.1 Rockfish models outperform Nanopolish and Megalodon for all probability thresholds. PR curves for other datasets are summarized in Supplementary Fig. 1.

### Rockfish improves CpG site-level methylation prediction

After evaluating Rockfish models using read-level examples, we proceeded to conduct a site-level evaluation in a similar manner. To begin, we aggregated the read-level predictions. Unlike the other methods, Rockfish employs a straightforward aggregation method without implementing additional filtering. In detail, we group the read-level Rockfish predictions by genome position and compute the proportion

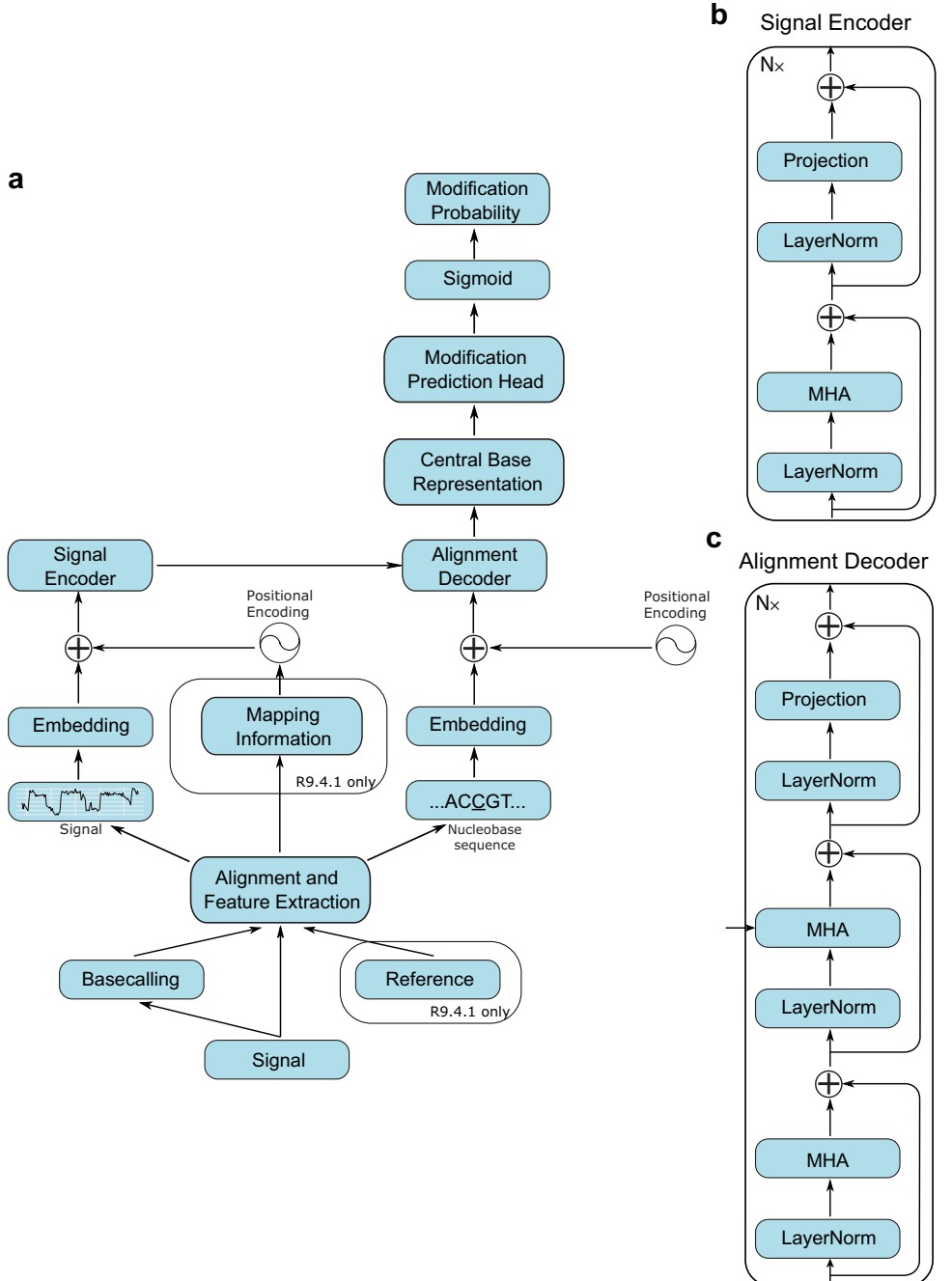

**Fig. 1 | Rockfish model. a** Overview of the Rockfish architecture. The nanopore signal is first basecalled using Guppy or Dorado. Next, signal, basecalled sequence (and reference for R9.4.1 models) are used for feature extraction. Each example consists of signal blocks, nucleobase sequence and mappings (only for R9.4.1). Signal blocks are embedded and processed using a signal encoder. Positional encoding for the embedded signal is given with the sine encodings and extracted mappings. Nucleobase subsequence is embedded and fed into the alignment decoder alongside contextualized signal representation. After decoding, the representation corresponding to the central base is fed into the modification prediction head. Modification probability is obtained by applying the sigmoid function. The architecture is described in detail in the Methods section. **b, c** Signal encoder and alignment decoder layout. They are defined as the standard Transformer (encoder-decoder) model with MHA denoting multi-head attention.

of methylated bases at a specific site. Megalodon, Nanopolish, and Remora exclude uncertain predictions, which leads to reduced methylation coverage. Furthermore, any site covered by fewer than 5 called reads from any ONT-based tool or WGBS was excluded from further analysis. This step was taken to ensure that the analyzed sites were adequately represented.

Figure 3a shows that the Rockfish small model outperforms other methods in the site level accuracy and F1 measure on all R9.4.1 datasets apart from H1ESc, where Megalodon Remora performs better. Results

are consistent for R10.4.1 where Rockfish outperforms Remora in both datasets. The models are evaluated across different genomic contexts and both R9.4.1 Rockfish models outperform other ONT-based tools in different genomic contexts with occasional exceptions for some R9.4.1 datasets. Results for all datasets (including additional R9.4.1 NA19240 and HX1) and different genomic contexts are summarized in Supplementary Data 2.

In the previous analysis, we evaluated only sites called by all methods. We now proceed by evaluating both the number of calls and

**a**

| | Tool | Accuracy | Precision | Recall | FPR | F1 |
|---|---|---|---|---|---|---|
| NA12878 (R9.4.1) | Megalodon Remora | 0.8804 | 0.8844 | 0.8797 | 0.1188 | 0.8820 |
| | Megalodon Rerio | 0.8878 | 0.9260 | 0.8470 | 0.0700 | 0.8847 |
| | Nanopolish | 0.8520 | 0.8635 | 0.8419 | 0.1375 | 0.8526 |
| | Rockfish base | **0.9393** | 0.9394 | 0.9412 | **0.0627** | **0.9403** |
| | Rockfish small | 0.9389 | 0.9392 | 0.9407 | 0.0629 | 0.9399 |
| H1ESC | Megalodon Remora | 0.8870 | 0.9796 | 0.8893 | 0.1295 | 0.9323 |
| | Megalodon Rerio | 0.8785 | 0.9929 | 0.8674 | 0.0436 | 0.9259 |
| | Nanopolish | 0.8489 | 0.9867 | 0.8386 | 0.0793 | 0.9066 |
| | Rockfish base | 0.9203 | **0.9961** | 0.9125 | **0.0249** | 0.9525 |
| | Rockfish small | **0.9224** | 0.9958 | **0.9152** | 0.0272 | **0.9538** |
| NA12878 (R10.4.1) | Remora | 0.9643 | 0.9692 | 0.9629 | 0.0342 | 0.9660 |
| | Rockfish | **0.9725** | **0.9756** | **0.9723** | **0.0271** | **0.9739** |

| | Tool | Accuracy | Precision | Recall | FPR | F1 |
|---|---|---|---|---|---|---|
| K562 | Megalodon Remora | 0.8958 | 0.5991 | 0.8971 | 0.1044 | 0.7185 |
| | Megalodon Rerio | 0.9433 | 0.7608 | 0.9009 | 0.0493 | 0.8249 |
| | Nanopolish | 0.8629 | 0.5231 | 0.8433 | 0.1337 | 0.6457 |
| | Rockfish base | **0.9574** | 0.8154 | 0.9207 | **0.0362** | **0.8649** |
| | Rockfish small | 0.9550 | 0.8035 | **0.9214** | 0.0392 | 0.8584 |
| Mouse (R9.4.1) | Megalodon Remora | 0.7850 | 0.9523 | 0.7477 | 0.1077 | 0.8377 |
| | Megalodon Rerio | 0.8537 | 0.9778 | 0.8215 | 0.0537 | 0.8928 |
| | Nanopolish | 0.8294 | 0.9645 | 0.7995 | 0.0846 | 0.8743 |
| | Rockfish base | 0.8813 | **0.9838** | 0.8540 | **0.0404** | 0.9143 |
| | Rockfish small | **0.8899** | 0.9821 | **0.8674** | 0.0455 | **0.9212** |
| Mouse (R10.4.1) | Remora | 0.8439 | 0.9527 | 0.7521 | 0.0452 | 0.8406 |
| | Rockfish | **0.8504** | **0.9667** | **0.7526** | **0.0313** | **0.8463** |

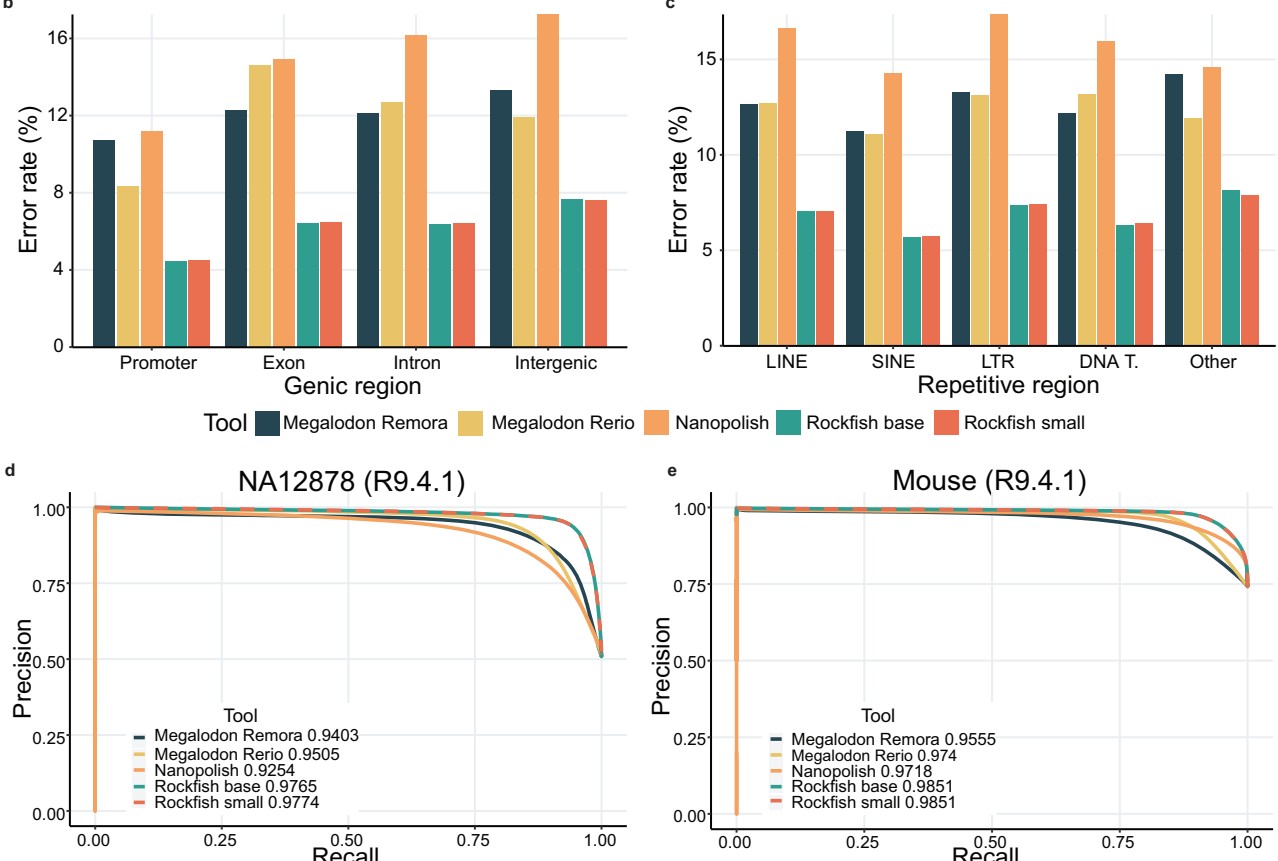

**Fig. 2 | Read-level evalution. a** A table presenting read-level evaluation for six different R9.4.1 and R10.4.1 datasets. For WGBS, only fully unmethylated or fully methylated sites with coverage higher than 5x were included. R9.4.1 Rockfish models significantly outperform Megalodon models and Nanopolish on all metrics and for all datasets. Similarly, R10.4.1 Rockfish model outperforms Remora on both datasets and in all metrics. The metrics are standardized with FPR denoting false positive rate, and the best scores bolded in the table. **b** The error rate for ONT-based tools in different genic regions (promoters, exons, introns and intergenic regions) on the R9.4.1 NA12878 dataset. R9.4.1 Rockfish models significantly increase accuracy for every genic region. **c** The error rate for ONT-based tools in different repetitive regions (LINE - long interspersed nuclear elements, SINE - short interspersed nuclear elements, LTR - long terminal repeats, DNA T. - DNA transposons and others) on the R9.4.1. NA12878 dataset. R9.4.1 Rockfish models significantly increase accuracy for every repetitive region. **d**, **e** show precision-recall (PR) curves for R9.4.1 NA12878 and C57BL/6 Neonatal mouse datasets that correspond to different species and contain balanced and imbalanced read-level methylation distribution. The average precision (AP) for each tool is given in the corresponding legend. Rockfish models significantly outperform Megalodon Rerio Megalodon Remora, and Nanopolish for all probability thresholds. Source data are provided as a Source Data file.

their concordance with bisulfite sequencing for Rockfish small, Megalodon Rerio, and Nanopolish in R9.4.1 and Rockfish base, and Remora in R10.4.1 datasets. Megalodon Rerio is chosen over Megalodon Remora since they achieve comparable results in 5 out of 6 datasets, but Megalodon Rerio outperforms Megalodon Remora on the R9.4.1 Mouse dataset. Since WGBS is a current state-of-the-art method, we choose it as the ground truth in this analysis. Figure 3b

describes relations between predicted positives (for methods based on ONT) and predicted positives achieved using WGBS for the R9.4.1 NA12878 dataset. Nanopolish calls notably fewer true positives (48.98% of all actual positives) than Rockfish (99.37%) and Megalodon (97.99%). Moreover, due to the aggressive filtering, Nanopolish calls the lowest number of false positives (0.43% of all actual negatives). Rockfish calls a slightly higher number of false positives (0.87%) than

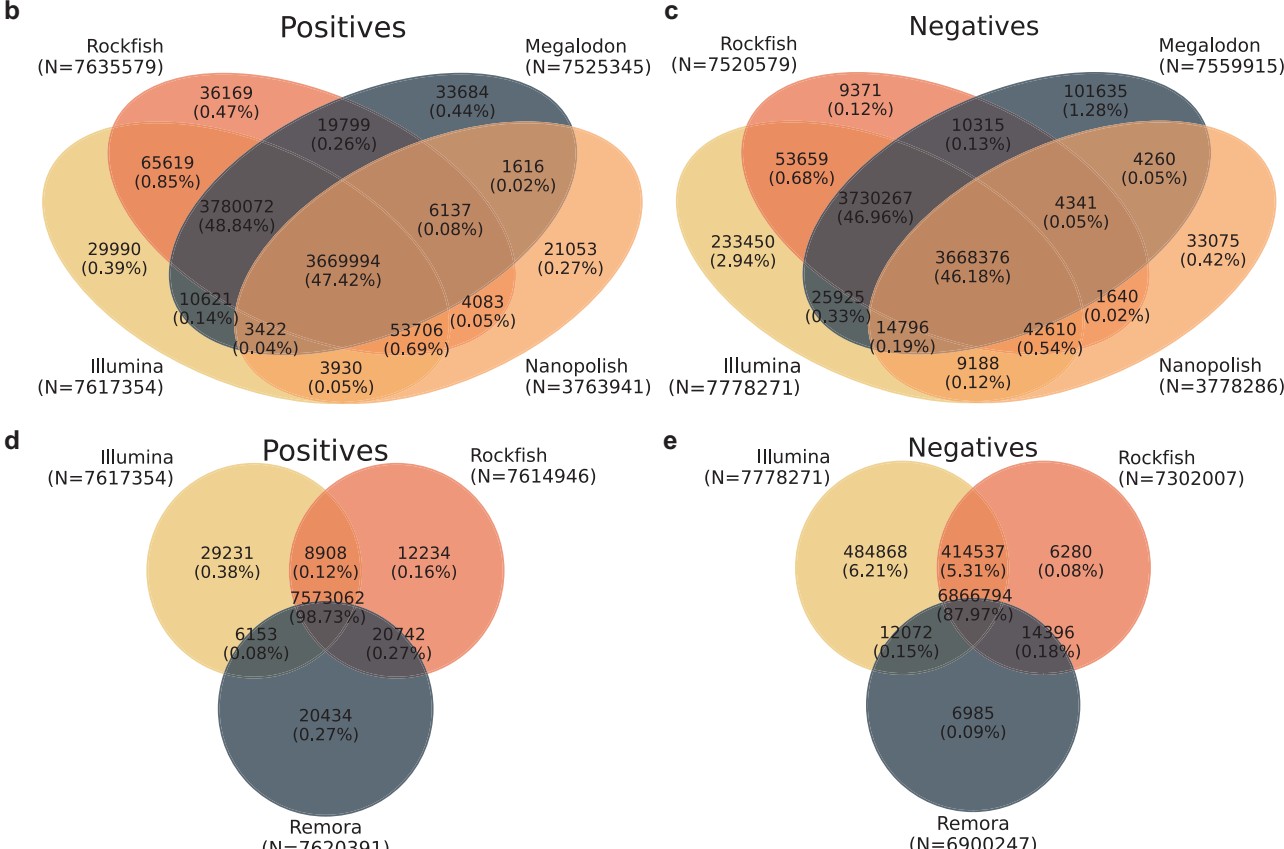

| | Tool | Accuracy | Precision | Recall | FPR | F1 |
|---|---|---|---|---|---|---|
| **NA12878** (R9.4.1) | Megalodon Remora | 0.9909 | 0.9910 | 0.9900 | 0.0091 | 0.9910 |
| | Megalodon Rerio | 0.9883 | **0.9924** | 0.9844 | **0.0077** | 0.9884 |
| | Nanopolish | 0.9905 | 0.9923 | 0.9888 | 0.0078 | 0.9905 |
| | Rockfish base | 0.9942 | 0.9917 | 0.9969 | 0.0085 | 0.9943 |
| | Rockfish small | **0.9945** | 0.9921 | **0.9970** | 0.0081 | **0.9946** |
| **H1ESC** | Megalodon Remora | **0.9951** | 0.9984 | **0.9960** | 0.0118 | **0.9972** |
| | Megalodon Rerio | 0.9929 | 0.9993 | 0.9927 | 0.0054 | 0.9960 |
| | Nanopolish | 0.9933 | 0.9987 | 0.9937 | 0.0098 | 0.9962 |
| | Rockfish base | 0.9931 | **0.9996** | 0.9926 | **0.0032** | 0.9961 |
| | Rockfish small | 0.9946 | 0.9995 | 0.9944 | 0.0039 | 0.9969 |
| **NA12878** (R10.4.1) | Remora | 0.9957 | 0.9946 | 0.9972 | 0.0059 | 0.9959 |
| | Rockfish | **0.9963** | **0.9957** | **0.9973** | **0.0047** | **0.9965** |

| | Tool | Accuracy | Precision | Recall | FPR | F1 |
|---|---|---|---|---|---|---|
| **K562** | Megalodon Remora | 0.9884 | 0.7361 | **1.0000** | 0.0119 | 0.8480 |
| | Megalodon Rerio | 0.9994 | 0.9815 | **1.0000** | 0.0006 | 0.9907 |
| | Nanopolish | 0.9945 | 0.8667 | 0.9811 | 0.0050 | 0.9204 |
| | Rockfish base | **1.0000** | **1.0000** | **1.0000** | **0.0000** | **1.0000** |
| | Rockfish small | **1.0000** | **1.0000** | **1.0000** | **0.0000** | **1.0000** |
| **Mouse** (R9.4.1) | Megalodon Remora | 0.9247 | 0.9938 | 0.9029 | 0.0155 | 0.9462 |
| | Megalodon Rerio | 0.9671 | 0.9951 | 0.9599 | 0.0130 | 0.9772 |
| | Nanopolish | 0.9731 | 0.9949 | 0.9683 | 0.0138 | 0.9814 |
| | Rockfish base | 0.9737 | **0.9963** | 0.9678 | **0.0100** | 0.9818 |
| | Rockfish small | **0.9788** | 0.9956 | **0.9754** | 0.0117 | **0.9854** |
| **Mouse** (R10.4.1) | Remora | 0.9830 | 0.9951 | 0.9640 | 0.0034 | 0.9793 |
| | Rockfish | **0.9862** | **0.9953** | **0.9716** | **0.0033** | **0.9833** |

**Fig. 3 | Site-level evaluation. a** A table presenting site-level evaluation for six different datasets. Only positions with coverage higher than 5x for each ONT-based tool and WGBS (whole genome bisulfite sequencing) were included. For the ground truth, we use only fully unmethylated or fully methylated positions concerning WGBS. R9.4.1 Rockfish models outperform Megalodon and Nanopolish on most metrics and for most datasets. Furthermore, R10.4.1 Rockfish model outperforms Remora on both evaluation datasets. The metrics are standardized with FPR denoting false positive rate, and the best scores bolded in the table. **b** positives and **c** negatives for different methods based on nanopore signal and the ground truth (Illumina) for the R9.4.1 NA12878 dataset. Rockfish is represented with the small model and Megalodon is represented with the Rerio model. Sample space is defined as the set of all fully unmethylated or methylated sites called by Illumina with at least 5x. Rockfish calls the highest number of true positives and true negatives and achieves high precision and recall. **d, e** show the same analysis for the R10.4.1 NA12878 dataset. Rockfish is represented with the base model and evaluated in all fully unmethylated or methylated sites called by Illumina with at least 5x. Rockfish calls the more true positives and negatives than Remora while achieving high precision and recall.

Megalodon (0.8%). Despite this marginally higher rate of false positives when compared to Megalodon, Rockfish is able to predict a notably larger number of positions, highlighting its enhanced detection capability. Figure 3c shows the distribution of predicted negatives with respect to bisulfite sequencing. Rockfish calls the highest number of true negatives (96.36% of all actual negatives) compared with Megalodon (95.64%) and Nanopolish (48.02%). Moreover, Rockfish calls a lower number of false negatives (0.33% of all actual positives) compared with Megalodon (1.55%) and Nanopolish (0.56%). Similar behavior occurred in the remaining R9.4.1 datasets. Figure 3d, e describe the same analysis for R10.4.1 NA12878 dataset. Rockfish calls slightly

more true positives (Rockfish 99.53% vs Remora 99.5%) while also calling less false positives (Rockfish 0.16% vs Remora 0.26% of all negatives). The difference is noticeably increased in the case of negatives where Rockfish calls significantly more true negatives (Rockfish 93.61% vs Remora 88.44%) while also calling slightly less false negatives (Rockfish 0.08% vs Remora 0.09% of all positives).

In addition, there are notably more positions called by Rockfish but not Megalodon that are supported by WGBS ground-truth calls (and Nanopolish) than those called by Megalodon but not Rockfish and supported by WGBS (and Nanopolish). There are 8.5x more true positive positions and 2.36x more true negative positions that are

called by Rockfish but not Megalodon than those called by Megalodon but not Rockfish in the R9.4.1 NA12878 dataset. This trend continues for all R9.4.1 evaluation datasets. The corresponding ratios for true positive positions in NA19240, H1ESc, and HX1 are 2.3, 2.08, and 1.84, respectively. The same ratios for true negative positions in NA19240, H1ESc, and HX1 are 11.31, 13.49, and 18.04, respectively. Similarly, there are 1.45x and 2.56x more true positive positions called by Rockfish but not Remora and supported by WGBS than vice versa in the R10.4.1 NA12878 and Mouse dataset, respectively. Finally, there are 34.34x and 69.78x more true negative positions called by Rockfish but not Remora and supported by WGBS than vice versa for the R104.1 NA12878 and Mouse.

### Effect of read-level predictions on site-level results

After noticing that Rockfish calls significantly more positions that are supported by WGBS than other ONT-based tools, and an overall larger number of positions, we proceeded to inspect differences in called positions overall and across different genomic contexts to understand the causes of such behavior. The complementary cumulative distribution function (CCDF) of the strand-specific coverage for Rockfish, Remora, and WGBS in the R10.4.1 NA12878 dataset is given in Fig. 4a. Remora achieves noticeably lower mean coverage (32.80x) compared with Rockfish (35.97x) due to the filtering of uncertain read-level calls. Read-level filtering is performed by determining the threshold for the 10% of the least confident calls from a subset of the reads and filtering all read-level calls with call confidence below that threshold[39]. We inspect the distribution of missing positions across genomic contexts to understand the mechanisms and effects of filtering uncertain calls. We define a missing position as one with coverage < 5 (as this is the value set as the threshold for evaluating the position). To understand the effects of Remora's filtering, we separated Remora's missing positions into two categories: positions where overall coverage (including methylation, canonical, and filtered calls) is < 5, and positions where valid coverage (including only methylation and canonical calls) is < 5, but overall coverage is ≥ 5. The positions with valid coverage < 5 but overall coverage ≥5 are of interest since they are removed from the site-level analysis due to low coverage, whereas that would not be the case if uncertain calls were not filtered. The distribution of missing position proportions in different genomic contexts is shown in Fig. 4b. It is visible that Remora filters more aggressively in specific genomic contexts such as promoters, islands, and GC-rich regions causing positions in those contexts to have valid coverage below the threshold more frequently. This results in a disproportionally higher number of missing positions compared with Rockfish. We proceed with the analysis by defining high-filtering positions (HFPs) as those with sufficient valid coverage (≥ 5) but above the expected number of filtered calls ( >10% of overall coverage) and identifying those positions across genomic contexts. The proportions of such positions across genomic contexts are shown in Fig. 4c demonstrating results similar to the missing positions analysis where some genomic contexts have noticeably higher frequencies of HFP. Finally, to understand the effects of filtering uncertain calls on methylation predictions, we analyze the site-level predictions of Remora and Rockfish in promoter regions. These regions were chosen for two reasons. First, Remora's filtering affects a significant number of promoter positions, and second, methylations in promotores play an important role in gene expression regulation. For this analysis, we define promoters as regions of 1000 base pairs ± TSS. Due to Remora's filtering exhibiting the strongest bias towards CpG islands (10.09% of CpG island positions are missing due to Remora filtering, and 43.36% of CpG island positions are high-filtering positions), we look into CpG rich and CpG poor promoter regions separately defining a promoter as CpG rich if the observed-to-expected CpG ratio is above 60%, and as CpG poor otherwise. We compare the results in all CpG poor and CpG rich promoter positions to those in high-filtering positions within CpG poor

and CpG promoters. The site-level results in promoter regions are shown in Fig. 4d. Remora performs worse in high-filtering positions compared to all positions (precision drops from 94.63% to 84.65% in CpG rich high-filtering positions) despite removing the most uncertain calls from those positions. Rockfish performs comparably in all positions and high-filtering positions having only a slight decrease in recall in CpG rich high-filtering positions (97.97% in all CpG rich promoter positions vs 95.58% in high-filtering CpG rich promoter positions). Furthermore, Rockfish outperforms Remora in all promoter positions with the difference between the two increasing in high-filtering positions. Rockfish achieves an F1 score of 99.56% compared with Remora's 98.99% in CpG poor high-filtering positions, and 96.9% compared with Remora's 90.96% in CpG rich high-filtering positions. Finally, we assess the concordance of methylation frequency predicted by ONT-based models and WGBS. Partially methylated CpG sites were also included in this analysis. Rockfish shows a higher correlation with WGBS compared to Remora in all promoter positions, with a significant difference in high-filtering positions where Rockfish achieves Pearson's $r$ of 0.9146 compared with Remora's 0.8871 in CpG poor high-filtering positions, and 0.8682 compared with Remora's 0.8403 in CpG rich high-filtering positions. Similar results are obtained for the R10.4.1 Mouse dataset and are shown in Supplementary Fig. 7. Full results for both R10.4.1 datasets are available in Supplementary Data 7–10.

### Methylation prediction results generated by rockfish models and WGBS are highly correlated

After evaluating Rockfish performance on read-level and site-level predictions, we further tested the correlation of site-level predictions with WGBS. In the subsequent experiments, we included partially methylated CpG sites to assess the concordance of methylation frequency predicted by ONT models and WGBS. Figure 5a demonstrates the correlation results for Megalodon models, Nanopolish and R9.4.1 Rockfish models on four R9.4.1 evaluation datasets. Rockfish models outperform Megalodon models and Nanopolish on all R9.4.1 datasets (Supplementary Data 3). Additionally, Rockfish models achieve higher correlation for most annotations, only the ones with lower GC contents (20% and 40%) being an occasional exception. The only dataset where Rockfish does not outperform other tools in higher GC content is K562. Rockfish and Remora achieve comparable correlation on both R10.4.1 datasets with Remora having a slightly higher correlation on R10.4.1 NA12878 and Rockfish having a slightly higher correlation on R10.4.1 Mouse dataset. Rockfish and Remora achieve comparable correlations across different genomic contexts. The largest differences are in "Other" and SINE repetitive regions where Remora achieves higher correlation, and CpG islands where Rockfish achieves higher correlation.

Figure 5b–d show the distribution of methylation frequencies for the R9.4.1 NA12878 dataset for (1) Rockfish base model and Rockfish small model, (2) Rockfish base model and WGBS and (3) Rockfish small model and WGBS. Methylation predictions obtained from the base and the small model exhibit a very high level of correlation (Pearson's $r = 0.9918$, $p = 0.0$). Moreover, the results from both the base model ($r = 0.9069$, $p = 0.0$) and the small model ($r = 0.9074$, $p = 0.0$) show high degrees of correlation with that of WGBS. In addition, the results given by Rockfish models are highly correlated with that of other ONT-based tools, especially with Megalodon Remora (Supplementary Fig. 5). The same pattern can be seen for other evaluation datasets (Supplementary Data 3).

Lastly, the methylation frequency was plotted with respect to the binned distance from the transcription start sites (TSSs) and the distributions of absolute differences between ONT-based tools (Fig. 5e) and WGBS for the R9.4.1 NA12878 dataset (Fig. 5f). Results from both R9.4.1 Rockfish models closely match that of WGBS (the median absolute difference is 0.00373 for the base model, 0.00263 for the small model), and stand lower than Megalodon Remora (0.01168),

 7

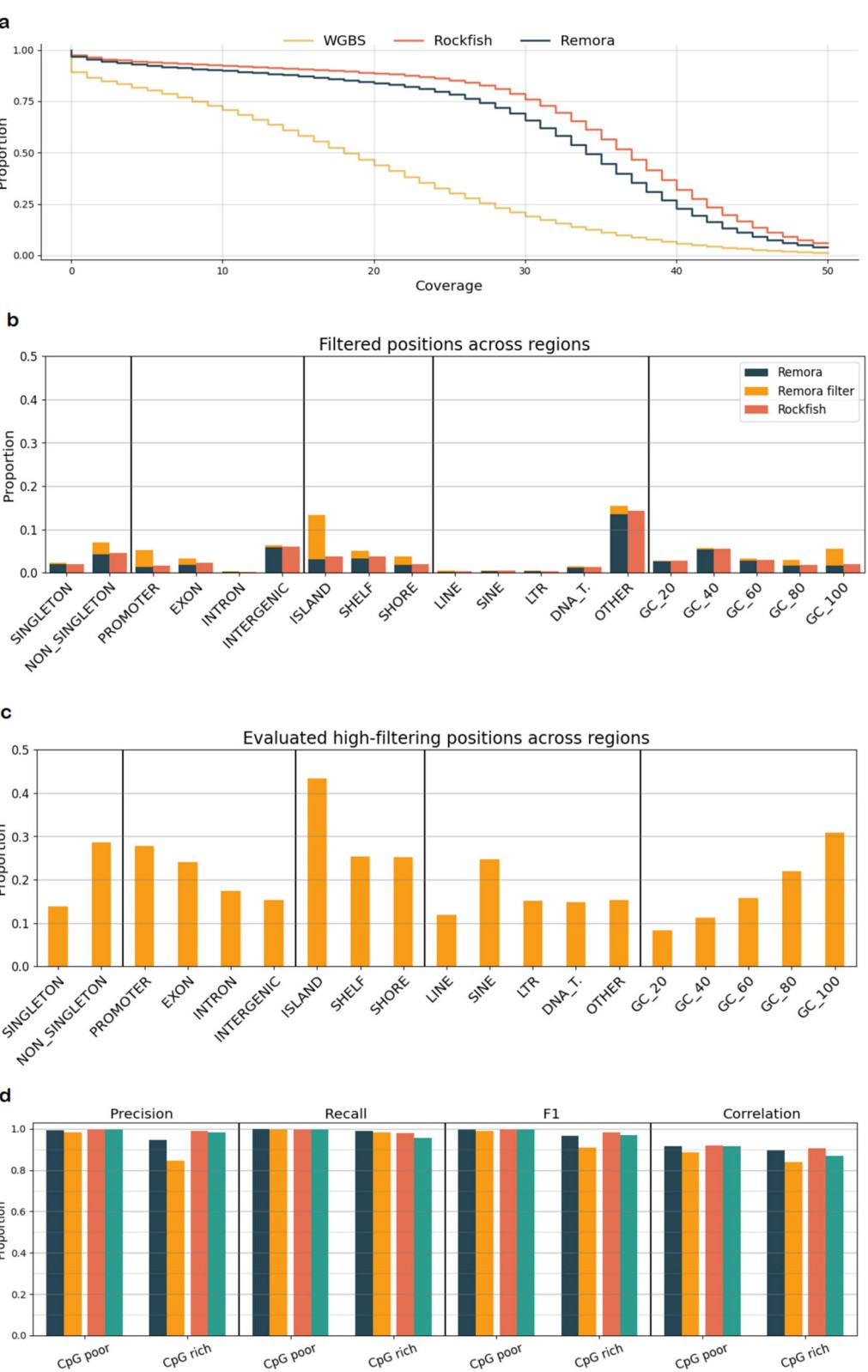

**Fig. 4 | Analysis of read-level predictions on site-level results. a** Complementary cumulative distribution function (CCDF) of the strand-specific calling coverage for each ONT-based method and whole genome bisulfite sequencing (WGBS) for the R10.4.1 NA12878 dataset. **b** Distribution of proportions of positions not evaluated due to low coverage with the distinction between positions with overall coverage below threshold (labeled as Remora) and positions with overall coverage above threshold but valid coverage below threshold due to Remora's filtering of uncertain calls (labeled as Remora filter) across different genomic contexts. **c** Distribution of proportions of high-filtering positions with sufficient valid coverage but above the expected number of filtered calls ( > 10%) across different genomic contexts. **d** Site-level evaluation in CpG poor and CpG rich promoter regions for all positions and high-filtering positions (HFP). Source data are provided as a Source Data file.

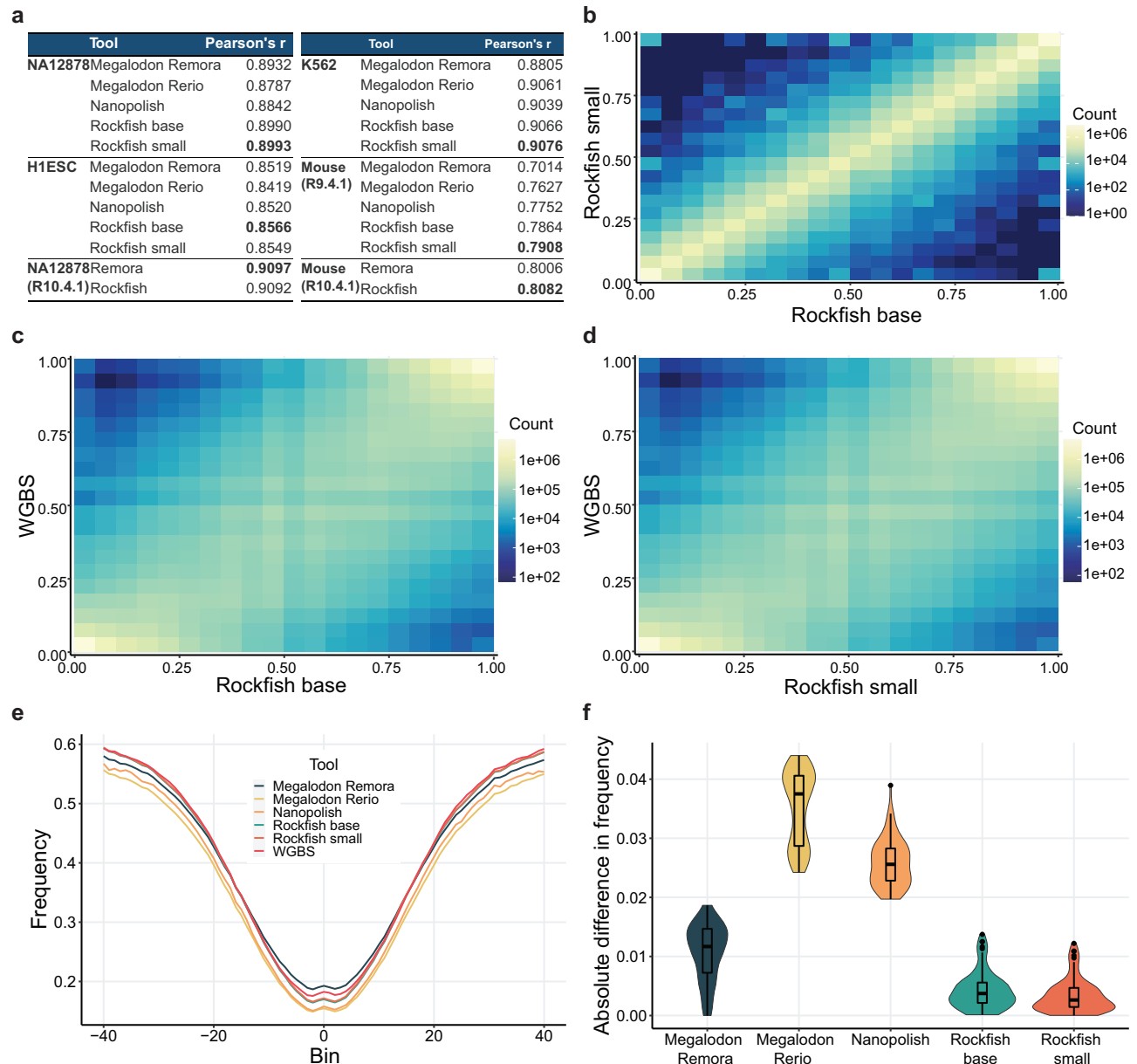

**Fig. 5 | Analysis of correlation between ONT-based tools and WGBS. a** A table presenting Pearson's correlation between the methods based on nanopore signal and whole genome bisulfite sequencing (WGBS). Only the sites with coverage higher than 5x for each tool were included. Both R9.4.1 Rockfish models outperform Megalodon and Nanopolish on every evaluation dataset. R10.4.1 Rockfish outperforms Remora on the Mouse dataset, but slightly underperforms on the NA12878 dataset. The highest correlation scores are bolded in the table. **b–d** 2D histograms representing correlation between **b** Rockfish models, **c** the base model and WGBS and **d** the small model and WGBS for the R9.4.1 NA12878 datasets. The models exhibit a high correlation with each other and with WGBS. Each axis is divided into 20 bins, and counts are represented on a log scale. **e** Methylation frequency for every ONT-based tool and WGBS with respect to the binned distance from the transcription start sites (TSSs) on the R9.4.1 NA12878 dataset. Both R9.4.1 Rockfish models show high consistency with other ONT-based tools and, more importantly, WGBS. **f** the distribution of the absolute difference between every ONT-based tool and WGBS. R9.4.1 Rockfish models reduce the absolute difference between ONT and WGBS. Data (n = 81) in the box plot are presented as follows: the centre line indicates the median, the bounds of the box represent the first and third quartiles (Q1 and Q3), and the whiskers extend to the minimum and maximum values within 1.5 times the interquartile range (IQR). Outliers beyond this range are plotted individually. Source data are provided as a Source Data file.

Nanopolish (0.02559) and Megalodon Rerio (0.03751) which has the highest distance from TSS based on the R9.4.1 NA12878 dataset. The same pattern can be observed for CTCF binding peaks in the R9.4.1 NA12878 dataset (Supplementary Fig. S6).

**Rockfish calls more sites compared with WGBS**

Next, we evaluated the number of Rockfish calls against that of WGBS for the R9.4.1 NA12878 dataset. The complementary cumulative distribution function (CCDF) of the strand-specific calling coverage for ONT-based methods and WGBS is given in Fig. 6a. The calling coverage is defined with respect to the outputs given by each tool. Since Rockfish does not perform any filtering of the read-level examples, it can call more CpG sites than Megalodon and Nanopolish for coverages lower than the sequencing coverage. Besides, both Rockfish models achieve the highest mean strand-specific calling coverage (~17x) compared with other ONT-based methods (Megalodon Remora ~14x, Megalodon Rerio ~14x, Nanopolish ~8x). Although WGBS was sequenced at much higher strand-specific coverage (~48x vs ~23x

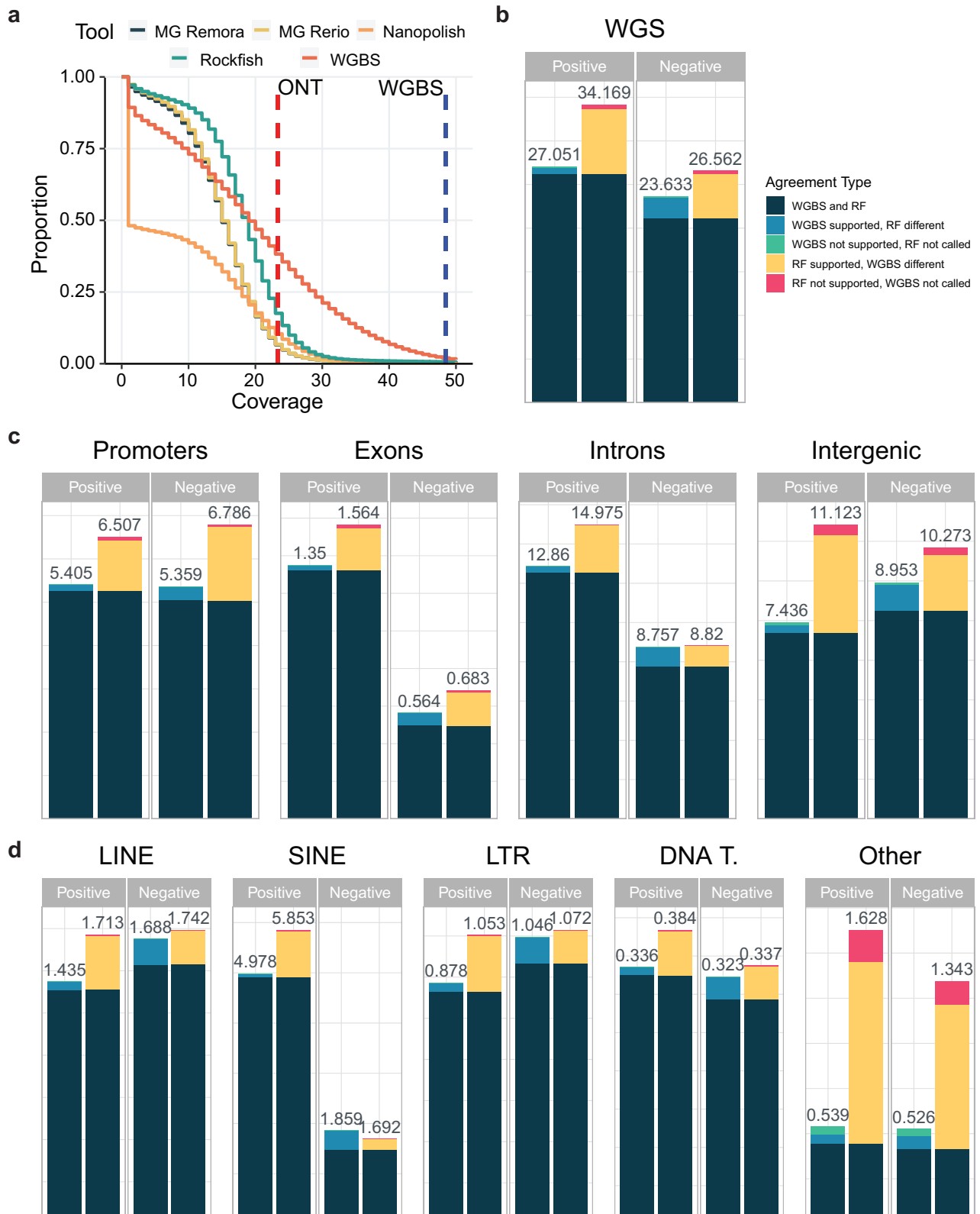

ONT), the mean strand-specific calling coverage (~18x) is close to that of Rockfish.

Moreover, we compared the number of highly confident positive and negative calls for WGBS and Rockfish. Highly confident calls were divided into three categories. The first category corresponds to the CpG sites for which both Rockfish and WGBS made concordant calls (both methods called a CpG site to be either positive or negative). In

the second category are the positions for which the calls for WGBS and Rockfish differ with the target method (either WGBS or Rockfish) having support from at least one ONT-based method. The last category reports the number of unique sites for which the tested method (WGBS or Rockfish) does not have any support and the other (Rockfish or WGBS) method does not produce a call. A CpG site is called positive if it is covered by at least five reads with a methylation frequency

**Fig. 6 | Analysis of number of called sites. a** Complementary cumulative distribution function (CCDF) of the strand-specific calling coverage for each ONT-based method and whole genome bisulfite sequencing (WGBS) for the NA12878 dataset. Vertical lines represent strand-specific sequencing coverage for ONT ( ~ 23x; red) and WGBS ( ~ 48x; blue). MG Remora denotes Megalodon Remora, while MG Rerio stands for Megalodon Rerio. **b** Stacked bar charts representing counts of highly confident positive and negative sites for WGBS (left) and Rockfish small (right; RF) for the R9.4.1 NA12878 dataset. A CpG site is defined as positive if the coverage is at least 5x and methylation frequency is at least 50%. A CpG site is defined as negative if the coverage is at least 5x and the frequency is less or equal to 50%. Sites with coverage less than 5x are labeled as not called. We define three

categories of highly confident sites: (1) WGBS and Rockfish are concordant, (2) WGBS and Rockfish calls differ with the target tool having support from at least one other ONT-based method, (3) WGBS without support and Rockfish without call and vice-versa. The numbers are given in millions ($10^6$). Rockfish calls more highly confident sites than WGBS on the whole genome. **c** Stacked bar charts representing counts of highly confident positive and negative sites for different genic and intergenic regions. Rockfish calls more highly confident sites for all genic and intergenic types. **d** Stacked bar charts representing counts of highly confident positive and negative sites for different repetitive regions. Except for unmethylated sites in SINE, Rockfish calls more highly confident sites than WGBS. Source data are provided as a Source Data file.

higher than 50%. A CpG site is called negative if it is covered by at least five reads with a frequency less than or equal to 50%. If a site was covered with less than five reads, it was deemed as uncalled. The results on the whole genome for the R9.4.1 NA12878 are given in Fig. 6b. Rockfish calls more highly confident positives and negatives (34.169M and 26.562M) compared with WGBS (27.051M and 23.633M). The majority of the calls for both WGBS and Rockfish were concordant calls (26.181M for positives, 21.179M for negatives). Rockfish had more support for both positive sites (7.468M) and negative sites (5.029M) compared with WGBS (0.77M and 2.364M, respectively). Lastly, Rockfish had more unique calls (positions where WGBS had no call) compared to WGBS unique calls (Rockfish small with no call) in both positive (0.52M vs. 0.1M) and negative sites (0.354M vs. 0.089M). We also explored the number of highly confident calls in genic and intergenic regions (Fig. 6c) and repetitive regions (Fig. 6d). Except for SINE, where WGBS calls more negatives (1.859M vs. 1.692M), Rockfish calls more highly confident positives and negatives for all types of annotations. The biggest difference between Rockfish and WGBS occurs in promoter regions (Rockfish calls 20.39% more positives and 26.63% more negatives), in intergenic regions (Rockfish calls 49.58% more positives and 14.74% more negatives) and in "Other" repeat types (Rockfish calls 202.04% more positives and 155.32% more negatives). Full counts are given in Supplementary Data 4.

## Assessment of Rockfish resource utilization

Besides the high prediction quality, other important aspects of a high-quality bioinformatics tool are the execution time and resource utilization. The K562 dataset was utilized to assess the running time, CPU, memory and GPU usage of the R9.4.1 ONT-based tools while R10.4.1 ONT-based tools were benchmarked on a subset of 400 000 reads sampled from R10.4.1 NA12878 dataset. Comprehensive information on Rockfish resource utilization can be found in the Supplementary Material.

## Discussion

In this paper, we present Rockfish, a deep-learning method for detecting 5mC at CpG sites in DNA at single-base, single-strand resolution. It utilizes a Transformer network, an architecture used in various visual, language and speech recognition tasks. Rockfish models do not rely on the explicit re-segmentation step. Rather, the signal information is combined with the reference subsequence and alignment information to obtain local alignment using multi-head attention. The ablation study showed an improvement in read-level classification of 2% in absolute value when sequence information is added.

Rockfish models are trained and evaluated on R9.4.1 and R10.4.1 datasets. For read-level evaluation, R9.4.1 Rockfish models significantly outperformed Megalodon models and Nanopolish on all six datasets. There is an increase in accuracy and F1 measure of up to 5 pp. Precision-recall curves show that the Rockfish models are resistant to class imbalance and are consistent across different species. Furthermore, the models outperform Megalodon and Nanopolish for all probability thresholds. R10.4.1 Rockfish model was evaluated against Remora on one human and one mouse dataset outperforming Remora

in both and proving not only consistency across different species but also adaptiveness of the architecture to newer versions of the nanopore. Significantly higher precision on the read-level prediction lowers the required coverage depth and reduces the costs for profiling the methylation landscape. Accurate read-level methylation prediction in long reads is crucial for the haplotype phasing[40], and aids haplotype-aware diploid assemblies[41]. Phasing facilitates potential applications such as cell type deconvolution in heterogeneous samples (e.g. blood or tumor samples[42]), as well as detecting allele-specific methylation typical for imprinting[43] and chromosome X inactivation[44] in homogenous samples.

Next, Rockfish models reduced the error rate for most datasets and most biological contexts for site-level predictions while including all called examples from the read-level prediction. Both R9.4.1 Rockfish models demonstrated a higher correlation with WGBS compared with Megalodon and Nanopolish. R10.4.1 Rockfish model demonstrated a comparable correlation with WGBS compared with Remora in both datasets. Furthermore, R9.4.1 Rockfish calls the highest number of true positives and negatives compared with Megalodon and Nanopolish while also calling fewer false positives and negatives compared with Megalodon, and even Nanopolish in some datasets (even though Nanopolish calls significantly fewer positions compared to Rockfish). There are also notably more true positive and true negative positions called by Rockfish but not Megalodon than those called by Megalodon but not Rockfish across all R9.4.1 evaluation datasets. Similarly, R10.4.1 Rockfish called a higher number of true positives and negatives than Remora while also calling less false positives and negatives. Furthermore, there are more true calls called by Rockfish and not Remora than vice versa, especially true negatives where Rockfish calls 34.34x more such positions than Remora. ONT-based methods had a smaller difference between sequencing and calling coverage than WGBS, possibly due to alignment ambiguity typical for short-read sequencing. Moreover, Rockfish had the highest mean strand-specific coverage compared with the other ONT-based tools and exhibited a higher number of highly confident calls than WGBS, especially for promoters, intergenic sites, and "Other" - repetitive regions excluding SINE, LINE, LTR, and DNA transposons. Although Rockfish calls a higher number of supported positions when compared to WGBS, the lower accuracy observed in lower GC contents and the comparison with WGBS (Fig. 5) raise the possibility of potential false positives. This aspect warrants further analysis and evaluation. The causes and effects of differences in coverages between ONT-based tools were further analyzed for R10.4.1 datasets by inspecting how filtering uncertain calls as a part of Remora's pipeline affects the coverage distribution and methylation prediction accuracy in different genomic contexts. Although Remora filters calls from the entire set based on a calculated confidence threshold, we noticed a bias towards filtering heavier in some genomic contexts. Namely, the proportion of positions affected by Remora's filtering is noticeably higher for CpG-rich regions such CpG islands and GC-rich regions. This is reflected in several ways. Firstly, there is a higher proportion of positions that are not evaluated due to low valid coverage (number of modified and canonical calls) which would not be the case if the uncertain calls were not filtered (the overall coverage is

sufficient). Secondly, the evaluated positions, despite having sufficient valid coverage, exhibit a higher proportion of high-filtering positions with the above-expected proportion of calls falling below the confidence threshold in those regions. Finally, despite filtering the most uncertain calls, Remora performs noticeably worse in high-filtering positions compared to results on all positions with precision drops of 9.97 pp and 6.92 pp in CpG rich promoters in R10.4.1 NA12878 and Mouse datasets, respectively. At the same time, Rockfish shows comparable performance when comparing all positions and those identified as high-filtering by Remora and significantly outperforms Remora in high-filtering positions, especially in CpG rich promoter context.

Subsequently, we evaluated the execution time and resource utilization of each ONT-based tool. Rockfish small significantly reduces the runtime compared to Rockfish base, achieving execution times comparable to Megalodon Remora and Nanopolish. Megalodon Rerio, although having the advantage of the shortest execution time, is typically trading canonical basecalling accuracy for the ability to simultaneously perform canonical and modified basecalling. This drawback led to the later decoupling of canonical and modified basecalling in subsequent ONT-based models[45].

It's worth highlighting two distinct advantages of Rockfish over other evaluated tools. Firstly, when it comes to read-level performance, Rockfish surpasses every other tool across all datasets. This increased accuracy allows for more precise phasing in de novo assembly and better detection of multiple cell types, a common occurrence in blood and cancer samples. Secondly, Rockfish exceeded the performance of all other tools with the mouse samples at both read and site levels. Notably, Rockfish base model was trained without any prior information about mouse data and genome and the small model was trained on soft-labeled mouse data. While we lack details about the training of other tools, these outcomes underscore the effectiveness of the transformer-based architecture, even on unseen datasets and genomes.

An intriguing observation we made is that when compared to all ONT-based tools, including Rockfish, the Illumina WGBS method additionally identifies a notably higher proportion of negative positions over positive ones (3% vs. 0.4% for R9.4.1 and 6.2% vs 0.4% for R10.4.1). We aim to delve deeper into this analysis in our upcoming research.

However, there's still potential for improvement. Currently, Rockfish models are trained using whole-genome bisulfite sequencing (WGBS), but they don't distinguish between 5mC and 5hmC methylation. This limitation stems from the absence of a high-quality ground-truth dataset for other modification types (e.g., oxBS for 5hmC) that could be paired with WGBS. Despite 5hmC being less common than 5mC, it would be advantageous to differentiate these methylations to gain deeper insights. The Rockfish pipeline can readily be adjusted to predict various modifications, whether in the same genomic context (CpG dinucleotides) or elsewhere.

While the Rockfish base model (with 12 encoder and 12 decoder layers) is highly accurate, it falls short in terms of computational efficiency compared to certain other tools. Additional architectural and engineering optimizations could lead to a decrease in computational time and resource usage.

The presented results demonstrate that Rockfish is a powerful and reliable method for extracting methylation information from the ONT raw signals. Further, the small model outperformed the base model on all datasets and required a shorter running time, showing the benefit of additional data and knowledge distillation.

5mC modifications are enriched in various genic elements and further relate to many biological phenomena, such as transcription regulation[46], chromatin architecture[47], diseases[48], ageing[49], memory formation[50], exercise[51] and many more. Therefore, the ability to detect 5mC modification at a single-base, single-strand resolution is critical for a deep understanding of DNA methylation's role in these biological phenomena. This knowledge might contribute to the early detection of disease onset, as well as patient stratification, treatment strategy

choice, and, in the future, even epigenome editing as a new direction of therapeutic targets.

Finally, due to its architecture, the Rockfish pipeline might be easily adapted to detect various types of DNA and RNA modifications.

## Methods

### Ethics statement
All animal protocols and experiments were approved by the Institutional Animal Care and Use Committee at the National University of Singapore. Handling of H1ESc was exempted from full review by the Institutional Review Board at the National University of Singapore.

### Statistics and reproducibility
Our study focuses on the development of a new computational method and does not involve biological or clinical samples. The statistical analyzes in our work are primarily related to the performance metrics of the algorithm, such as accuracy and F1 score, rather than traditional statistical methods used in experimental research. No statistical method was used to predetermine sample size. No data were excluded from the analyzes. The experiments were not randomized. The Investigators were not blinded to allocation during experiments and outcome assessment.

### Feature extraction
The processing of the sequenced data begins with basecalling. Data obtained from R9.4.1 flowcells was stored as FAST5 files and basecalled using Guppy basecaller (https://nanoporetech.com/nanopore-sequencing-data-analysis) (version 5.0.14). Similarly, data from R10.4.1 flowcells was stored in POD5 format and basecalled using Dorado (https://github.com/nanoporetech/dorado) (version 0.4.2). For both chemistry types, we used a super-accurate DNA basecalling model to ensure optimal accuracy. Besides obtaining the basecalled sequences, we also generate move tables from both basecallers. The move table defines sequence-to-signal alignment that is used for obtaining nanopore signal corresponding to the subsequence of interest. Examples of Guppy and Dorado commands, including all input arguments, are provided in the Supplementary Material.

After obtaining sequence-to-signal alignment, for R9.4.1 data, basecalled ONT reads are aligned to a reference genome using mappy, minimap2 biding for Python[52]. The Mappy version used for the sequence alignment is 2.24. The reference genome used for human datasets is T2T-CHM13 (v2.0), the complete assembly of a human genome[53]. We use Genome Reference Consortium Mouse Build 39 (GRCm39) for mouse datasets. We choose the best primary alignment and generate reference-to-query mapping by parsing the CIGAR string. Unmapped reads are not processed further. For R10.4.1 data, we skip the reference alignment step and directly utilize the basecalled sequences as input for the decoder model. In subsequent sections, the term 'nucleobase (sub) sequence' will refer to the reference (sub)sequence in the context of R9.4.1 data, and to the basecalled sequence for R10.4.1 data.

Next, we find relevant positions - i.e., CpG dinucleotides in the nucleobase sequence and extract a subsequence of length $l = 31$ around each relevant position (cytosine in the CpG context). For each subsequence, we extract corresponding $p$ signal points. We divide the signal into signal blocks $\mathbf{B} \in \mathbb{R}^{s \times b}$ where each block corresponds to $b$ points, $s = p/b$. The parameter $b$ is determined by a particular basecalling model. For 4 kHz data and the super-accurate model $b = 5$, while for 5 kHz data $b = 6$. For quality and computational reasons, examples with less than $0.5s$ or more than $5s$ blocks are discarded. Moreover, for R9.4.1 models, we extract relative reference indices $\mathbf{r} \in \mathbb{N}_0^s$ and relative query indices $\mathbf{q} \in \mathbb{N}_0^s$. The relative reference indices, indicated as $\mathbf{r}$, determine the specific base within the subsequence that a signal block corresponds to. This relationship is established through a transitive mapping that links the signal to the reference. This transitive mapping is defined by combining the move table, which maps the signal to the

basecalled sequence, with the minimap2 alignment, which maps the basecalled sequence to the reference. Conversely, the relative query indices $\mathbf{q}$ indicate the particular basecalled base that a signal block is connected to. Both $\mathbf{r}$ and $\mathbf{q}$ will be later used as positional encodings in R9.4.1 models.

## Architecture

The Rockfish architecture consists of four components: signal projection layer, nucleobase sequence embedding layer, Transformer and modification prediction head. First, we use a linear projection layer $\mathbf{W}^S \in \mathbb{R}^{b \times f}$ to transform signal blocks into a local representation $\mathbf{S} \in \mathbb{R}^{s \times f}$, $\mathbf{S} = \mathbf{BW}^S$. Here, parameter $f$ is the dimension of both signal and reference latent space.

Next, we add positional encodings to the signal representation since the Transformer architecture itself is not processing data sequentially. For R9.4.1 models, we add four types of positional encoding: both cosine and sine encodings of the absolute position, the cosine of the relative reference indices and the cosine of the relative query indices. Alongside traditionally used cosine and sine encodings[54], we decided to use both reference and query indices to provide additional positional information to the model. For R10.4.1 model we use only cosine and sine encodings. The final positional encoding matrix $\mathbf{P} \in \mathbb{R}^{s \times f}$ for a specific example for R9.4.1 model (left) or R10.4.1 model (right) is given with:

$$
P_{ij}^{R9} = \begin{cases} \cos(i \cdot \text{pt}(j)) & \text{if } j \bmod 4 = 0, \\ \sin(i \cdot \text{pt}(j)) & \text{if } j \bmod 4 = 1, \\ \cos(\mathbf{r}_i \cdot \text{pt}(j)) & \text{if } j \bmod 4 = 2, \\ \cos(\mathbf{q}_i \cdot \text{pt}(j)) & \text{otherwise} \end{cases}
$$
$$
P_{ij}^{R10} = \begin{cases} \cos(i \cdot \text{pt}(j)) & \text{if } j \bmod 2 = 0, \\ \sin(i \cdot \text{pt}(j)) & \text{otherwise} \end{cases} \tag{1}
$$

Positional term function $pt$ is defined as $\text{pt}(i) = 10000^{-4\lfloor i/4 \rfloor / d}$. The final input to the Transformer encoder is obtained by adding positional encodings $P$ to the signal representation $S$.

We use a Transformer encoder[54] to capture contextual information between the signal points by iteratively updating signal representation $\mathbf{S}$. To improve training stability, the original (Post-LN) encoder layer is replaced with the Pre-LN layer[55]. First, inputs to the encoder layer are normalized using layer normalization[56]. Next, we update the signal representation by using multi-head attention (MHA):

$$
\text{Attention}(\mathbf{K},\mathbf{Q},\mathbf{V}) = \text{softmax}\left(\frac{\mathbf{QK}^\top}{\sqrt{d}}\right)\mathbf{V} \tag{2}
$$

$$
\text{MHA}(\mathbf{K},\mathbf{Q},\mathbf{V}) = [\text{Attention}_1, \ldots, \text{Attention}_H]\mathbf{W}^O. \tag{3}
$$

where $[\cdot]$ is the concatenation operator, $H$ is the number of heads, $d = f/H$ and $\mathbf{W}^O \in \mathbb{R}^{f \times f}$ is an output projection matrix. Scaled dot-product attention for each head is given as $\text{Attention}_i = \text{Attention}(\mathbf{KW}_i^K, \mathbf{QW}_i^Q, \mathbf{VW}_i^V)$ where $\mathbf{W}_i^K, \mathbf{W}_i^Q, \mathbf{W}_i^V \in \mathbb{R}^{f \times d}$ are projection matrices. Finally, the result of the multi-head attention is added to the input.

In encoder multi-head attention is defined as self-attention since matrices $\mathbf{K}$, $\mathbf{Q}$ and $\mathbf{V}$ all correspond to the normalized input. After multi-head attention, signal representation is again normalized using layer normalization and fed to the linear projection layer. Linear projection layer is a simple two-layer feed-forward network given as: $\text{Projection}(\mathbf{x}) = \mathbf{W}^{out} \text{GELU}(\mathbf{W}^{in}\mathbf{x} + \mathbf{b}^{in}) + \mathbf{b}^{out}$ where GELU is a Gaussian Error Linear Unit[57], $W^{in} \in \mathbb{R}^{f \times p}$, $\mathbf{b}^{in} \in \mathbb{R}^p$, $W^{out} \in \mathbb{R}^{p \times f}$ and $\mathbf{b}^{out} \in \mathbb{R}^f$. The final output is given by adding projection output to the projection input. This process is repeated $L_{enc}$ times.

After obtaining contextualized signal representation, we embed nucleobase subsequence using a simple look-up table $\mathbf{E}^B \in \mathbb{R}^{6 \times f}$ to obtain localized sequence representation $\mathbf{R} \in \mathbb{R}^{l \times f}$. Except for four canonical bases, we also define two extra tokens: unknown token and mask token. Unknown token [UNK] represents all non-canonical bases with respect to the FASTA format. Mask token [MASK] is used during training for the base prediction task. Next, positional encodings (same as in[54]) are added to the localized sequence representation.

We use the Pre-LN Transformer decoder to obtain contextualized sequence representation. In the decoder, the starting sequence representation is iteratively updated using decoder layers. Each layer uses both current sequence and contextualized signal representations to update the sequence representation. First, the current representation is normalized by applying layer normalization. Next, the representation update is performed using self-attention, the same as in the encoder layer. The output from self-attention is added to the input. The resulting representation is then normalized and passed to the multi-head attention where query matrix $\mathbf{Q}$ corresponds to the current representation and matrices $\mathbf{K}$ and $\mathbf{V}$ correspond to the contextualized signal representation. The motivation for using multi-head attention is to learn the alignment between signal and reference sequence and to update sequence representation with relevant signal information. Next, MHA output is added to the input, normalized and then passed through the projection layer and added to the input, the same as in the encoder. We repeat this process $L_{dec}$ times to obtain the final contextualized sequence representation. More details regarding Transformer architecture and the pseudocode can be found in[58].

To obtain modification probability, we take the contextualized representation corresponding to the central cytosine $\mathbf{B}^c_{\lfloor l/2 \rfloor} \in \mathbb{R}^f$ and pass it through the modification prediction head. The modification prediction head is a linear layer that outputs unnormalized modification probability.

The hyperparameters for all Rockfish models are listed in Supplementary Data 6. Base and small models share most of the hyperparameters other than hyperparameters determining the number of parameters in the model. Rockfish small, therefore, has 6 instead of 12 encoder and decoder layers, 1024-dimensional instead of 2048-dimensional feed-forward output, and 128-dimensional features instead of 384-dimensional ones. All hyperparameters were determined empirically.

## Training and evaluation

**Modification prediction task.** We model modification prediction as a binary classification task. Loss for $i$-th example is given with: $\mathcal{L}_{mod}^{(i)} = \text{BCE}(z^{(i)}, y^{(i)})$ where BCE is binary cross-entropy loss, $z^{(i)}$ is unnormalized probability, $y^{(i)}$ is the ground truth.

**Auxiliary tasks.** To improve learning and generalization, we implement two auxiliary tasks during training: base prediction task and signal classification task. Both base prediction and signal classification tasks are related to masking, a self-supervision technique used during model pre-training[59,60] or model training[61]. In every iteration, for the base prediction task, we randomly choose $p_{mask}$ of all reference bases and mask them using the mask token. Moreover, we also randomly flip $p_{flip}$ of all bases and choose the new base with equal probability. During training, contextualized representations corresponding to the masked and flipped positions are passed to the base prediction head to obtain logits for each base. These logits will be used to predict the correct bases with cross-entropy loss $\mathcal{L}_{bases}$. The base prediction task forces the model to learn to predict the correct reference bases. This helps the model to learn the local signal-to-reference alignment and correct any errors introduced during sequencing, basecalling and alignment. Moreover, it helps the model to learn the alignment between the signal and reference sequence.

The second auxiliary task is the signal classification task. The idea of the signal classification task is to force the model to learn the context for each signal block independent of the specific task. Since signal blocks are continuous, we introduce representative vectors, named codewords, to be used for classification. The collection of codewords is a codebook $C \in \mathbb{R}^{f \times K}$, $K$ being the number of different codewords. First, we randomly choose $p_{signal}$ of all the signal blocks $S$ that will be masked. Next, we calculate the probability of a local signal representation belonging to $i$-th class: $P_l(i|b) = e^{c_i b} / \sum_j e^{c_j b}$. The target class for a specific signal block is given with $t = argmax_i P_l(i|b)$. Elements of masked blocks, relative reference and query indices are all set to zero. After a forward pass through the Transformer encoder, we calculate the probability of contextualized signal representation belonging to $i$-th class in the same way as above and use these probabilities as predictions for cross-entropy loss $\mathcal{L}_{signal}$. To ensure non-trivial solution, we introduce diversity loss $\mathcal{L}_{diversity} = -\sum_i u \log(u) + \sum_i \bar{p}_i \log(\bar{p}_i)$ where $u = 1/K$ and $\bar{p}_i$ is the average hard probability of $i$-th class calculated for each batch.

**Final loss and optimization.** The total loss for the training is given as a linear combination of all losses:

$$\mathcal{L} = \mathcal{L}_{mod} + \alpha * (\mathcal{L}_{bases} + \mathcal{L}_{signal} + \mathcal{L}_{diversity}) \qquad (4)$$

where $\alpha$ is a scaling parameter. All weights are optimized using a modified version of Adam[62] which decouples weight decay from the optimization procedure[63]. In our experiments, the learning rate is set to $3e^{-4}$ and weight decay is set to $1e^{-4}$. Running average coefficients were set at their default values $\beta_1 = 0.9$, $\beta_2 = 0.999$.

**Evaluation.** We performed read-level, site-level and correlation evaluations to compare our method against other ONT methods. For read-level and site-level we reported accuracy, precision, recall, false-positive rate (FPR) and F1 score for each ONT-based tool. Moreover, we plotted the precision-recall curve for read-level predictions. Venn diagrams were used to describe the relations between site-level predictions for all ONT methods and WGBS. All evaluation metrics were calculated using their standard definitions. Average precision was calculated as defined in[64]. For human data, the evaluation was performed on chromosomes 1-22, X and Y if the data corresponds to the male genome. For mouse data, all chromosomes were used.

We used WGBS (or RRBS) data as the ground truth. The pipeline used for processing WGBS data includes Trim Galore (https://github.com/FelixKrueger/TrimGalore) used for adapter and quality trimming and Bismark[65] used for alignment, deduplication and methylation extraction. The pipeline is fully described in the Supplementary Material. To reduce bias related to the misaligned Illumina reads we defined lower and upper coverage bounds. For read-level and site-level evaluation, we excluded genomic positions which have coverage less than $max(P_5, 5)$ where $P_5$ is the 5-th percentile for coverage distribution in the corresponding WGBS. For correlation evaluation, we also put an upper bound to be $P_{95}$ (95-th percentile). Moreover, for read-level and site-level evaluation, we removed all positions that are partially methylated. A CpG site is defined to be partially methylated if the frequency in bisulfite sequencing is between 0.01 and 0.99.

For read-level evaluation, we evaluated only examples that are present in all ONT-based tools. For site-level and correlation evaluation, we used only positions with ONT coverage higher than 5x.

Moreover, for human datasets, we reported read-level, site-level and correlation results for different types of annotations: gene annotations, repetitive regions, CpG islands, GC content, and CpG count, similarly as in[34].

For gene annotations, we defined four categories: promoters, exons, introns and intergenic positions. Annotations for transcription start sites (TSS), genes and exons were extracted from the annotation file. Examples were labeled as promoters if they belong to a region ± 2000 around the TSS. Positions corresponding to the introns were obtained by subtracting exons from genes using intersectBed[66]. All other positions were labeled as intergenic positions. If a position was labeled with multiple annotations, we defined the following precedents: promoter > exon > intron > intergenic.

Annotations for repetitive regions were generated using Repeat-Masker. We defined five repeat categories: SINE, LINE, LTR, DNA transposons and "Other" (positions belonging to Simple, Low complexity, Satellite, RNA, Others or Unknown repeat class).

For the analysis of CpG islands, we defined three categories: CpG islands, CpG shores, and CpG shelves. First, regions corresponding to CpG islands were extracted from the CpG islands annotation file. Regions 2000 bp to the left and right of each CpG island were labeled as CpG shores. Regions 2000 bp to the left and right of each CpG shore were labeled as CpG shelves. GC content corresponds to the frequency of cytosines and guanines in 5-base windows. For CpG count, we defined two categories: singleton and non-singleton CpG. The positions were labeled as singletons if there was only one CpG (central) in the 20-bp region around the central CpG. Otherwise, a position was labeled as a non-singleton.

Furthermore, we plotted methylation frequencies with respect to the binned distance to TSS and CTCF binding peaks for the NA12878 dataset. For each TSS, we calculated the corresponding bin according to $bin = (pos - P + \lfloor B/2 \rfloor)/B$ where $pos$ is the tested position, $P$ is the TSS and $B$ is the bin size. CTCF binding peaks were obtained by running ENCODE chip-seq-pipeline2 (https://github.com/ENCODE-DCC/chip-seq-pipeline2). Excluded regions for the T2T-CHM13 genome were obtained from[67]. For TSS evaluation we set bin size to $B = 50$ base-pairs and for CTCF binding peak evaluation to $B = 125$, same as in[34]. In these experiments, we did not perform an intersection between tools, but we filtered out positions individually. Positions were discarded if coverage was less than 3x for ONT data and 5x for WGBS data. The methylation frequency for each bin was calculated by averaging frequencies for positions assigned to the given bin.

Moreover, we compared the coverages and the number of calls for Rockfish and WGBS. We calculated strand-specific sequencing coverage for both ONT and WGBS by counting the total number of sequenced bases and dividing the number by two times the size of the human genome (3.117 Gbps) $cov = n_{bases}/(2 \times 3.117 \times 10^9)$. A complementary cumulative probability distribution for a specific coverage is defined as the proportion of CpG sites with equal or higher strand-specific calling coverage divided by all CpG sites $ccdf(cov) = N_{\geq cov}/N_{sites}, \forall cov \in \mathbb{N}_0$. The calling coverage is the coverage provided by the final output for each tool. We define three types of highly confident CpG positive and negative sites: (1) sites with the calls concordant between both WGBS and Rockfish, (2) sites with the calls discordant between WGBS and Rockfish with the target method (WGBS or Rockfish) being supported either by Megalodon and/or Nanopolish and (3) sites for which either Rockfish or WGBS does not produce any call. A CpG site is defined to be positive if the strand-specific coverage is at least 5x with more than 50% methylation frequency. A CpG site is labeled as negative if the strand-specific coverage is at least 5x and methylation frequency is less or equal to 50%. A CpG site is deemed uncalled if coverage is less than 5x.

Lastly, we plotted the execution time and resource utilization for every ONT-based tool. Experiments were repeated three times, with the bar sizes representing the average running time for each step. Experiments involving R9.4.1 methods were run on NVIDIA DGX-1, with the following configuration: 2x 20-core Intel Xeon E5-2698 v4 2.2 GHz CPUs, 512 GB 2133 MHz DDR4 LRDIMM RAM, 8x NVIDIA V100 32 GB VRAM GPUs with 4x 1.92 TB SSD RAID 0 storage. We limit each tool to 32 threads or processes and one GPU. The R10.4.1 model was optimized using FlashAttention2[68], which is an optimized version of the attention mechanism. Experiments for R10.4.1 methods were run on

the cluster server with the following configuration: AMD EPYC 7713P 64-Core Processor, 1 TB RAM, 4x NVIDIA A100 80 GB VRAM GPUs. Execution time was measured using the GNU *time* utility (version 1.7). GPU memory consumption and GPU utilization were computed from the output of the NVIDIA System Management Interface, commonly referred to as *nvidia − smi* (v470.141.10).

Rockfish R9.4.1 models were compared with three ONT-based methods, Megalodon (version 2.4.2) coupled with Remora (dna_r9.4.1_e8 sup 0.0.0 5mc, v0.1.2), Megalodon coupled with Rerio (res_dna_r941_min_modbases_5mC_CpG_v001) and Nanopolish (0.14.0). Guppy (5.0.14) super accurate model was used as the canonical basecalling backend. Internally, Megalodon uses minimap2 (2.24; via Mappy) for pairwise alignment. For Nanopolish, reads were basecalled with Guppy (5.0.14) and aligned with minimap2 (2.24). Rockfish R10.4.1 model was compared against Remora (dna_r10.4.1_-e8.2_400bps_sup@v4.2.0 5mCG_5hmCG) which is integrated into Dorado basecaller (version 0.4.2). For the WGBS methylation pipeline, we used Trim galore (0.6.7) to perform quality and adapter trimming. After trimming, reads were processed using Bismark (0.23.1). More details about the tools and examples of commands used for evaluation can be found in Supplementary Material.

**Ablation study.** The ablation experiments were run using the Rockfish base model with a reduced number of layers due to computational constraints. The model consisted of 4 encoder and 4 decoder layers. The ablation was performed using the original training dataset and trained until convergence. The obtained models were evaluated on the ONT NA24385 chromosome 1 dataset. The results of the study show the importance of the alignment decoder component (accuracy was up by 2% for prediction on the read-level). More details can be found in supplementary information and Supplementary Fig. 7.

**Datasets**

**Mouse datasets.** Three mouse samples were used for the knowledge distillation training. C57BL/6 mice were bred and maintained under standard 12:12 h light/dark conditions at the National University of Singapore. The mice for cardiomyocyte isolation experiments were maintained at standard conditions. The mice for diet control were maintained in the following manner: co-housed male mice from mixed litters (n = 5 per cage) were initially provided with standard chow diet (2018, 18% Protein Rodent Diet, Envigo) until weaning (starting from 3-4 weeks of age after birth), then the diet for HFHS group was switched to purified high fat/high sugar (sucrose) diet (45% fat DIO diet, TD.08811, Envigo) ad libitum. Water was also provided ad libitum. Eight weeks into the study, the blood was drawn and processed as described below.

Blood from the facial submandibular vein was collected in EDTA-coated micro-containers (BD, 365974). Following immediate centrifugation, at 4 °C for 15 min at 2000 x g, blood cell pellets were collected and stored at −80 °C until analysis. Upon thawing, the red blood cell was burst by RBC Lysis Buffer: 100 mM Tris, pH 7.5 (1st Base, BUF-1416-1L-pH7.5), 0.2 mM EDTA (Invitrogen, 15575020) buffer[69,70]. Then the genomic DNA (gDNA) in the buffy coat was extracted as described below.

The neonatal and adult mouse cardiomyocyte isolation was carried out following the previous literature[71,72].

**H1ESc datasets.** H1 human embryonic stem cells (WiCell, WA01, hPSCReg ID WAe001-A) were maintained in mTeSR (Stemcell Technologies, 85850) on growth factor-reduced Geltrex (1:200 dilution, Thermo fisher, A1413202) coated plates at 37 °C with 5% $CO_2$. Cells were dissociated using ReLeSR (Stemcell Technologies, 05872) for gDNA extraction.

For the native dataset, cell pellets were resuspended in 10 mM Tris pH 7.5 buffer and digested with 400 $\mu$g RNase A (Thermo Scientific,

EN0531) at 37 °C for 30 mins. Then 0.5% SDS (final concentration, 1st Base, BUF-2051-1L) and 600 $\mu$g Proteinase K (Invitrogen, 4333793) were added and incubated at 50 °C for 3 hours for digestion. The reaction mixture was purified by phenol-chloroform extraction following standard protocol, with UltraPureTM Phenol:Chloroform: Isoamyl Alcohol (Invitrogen, 15593031) followed by one-time chloroform extraction to remove residual phenol. Then the native gDNA was precipitated by 2.5 volume of absolute ethanol and 10% volume of 3 M sodium acetate (pH 5.2, Thermo Scientific, R1181) following standard protocol and washed once with 70% ethanol. The pellet was dried for 5 minutes and slowly hydrated at 4 °C, in 10 mM Tris pH 7.5 buffer for more than 72 hours. The hydrated native gDNA extracted was quantified and quality-checked by two independent researchers, and the library prep and sequencing were carried out by the Integrated Genomics Platform of the Genome Institute of Singapore.

To generate the negative control, modifications on the native genomic DNA were wiped out using the REPLI-g Mini Kit (Qiagen, 150023). Subsequently, the positive control was produced by treating the whole genome amplified (WGA) sample (i.e., the negative control) with M.SssI methyltransferase (NEB, M0226S), as per the recommended conditions. Both the negative and positive control reaction mixtures underwent the same purification and treatment protocols from this point onwards. They were digested with Proteinase K at 50 °C for 30 minutes and purified by phenol-chloroform extraction, following standard protocols stated in the previous section. It was subsequently washed once with 70% ethanol. The resultant pellet was dried for 5 minutes and slowly hydrated at 4 °C in 10 mM Tris pH 7.5 buffer for over 72 hours. The resulting DNA fragments were quantified and quality-checked using Qubit, Nanodrop, and the Agilent Genomic DNA ScreenTape (5067-5365) on a 2200 TapeStation system. The resultant DNA was then prepared for the library and sequenced on a MinION R9.4.1 (FLO-MIN106), adhering to the official library prep ligation protocol for SQK-LSK109 and standard sequencing parameters.

**WGBS.** Both mouse and human data follow the same procedure for WGBS preparation and sequencing. The DNA used in whole genome bisulfite sequencing (WGBS) was first sheared by Covaris S2 (Covaris, USA) at 10% duty cycle, 2 × 40 s, intensity 5, cycle per burst 200 at 100 $\mu$L. Resulting DNA fragments were library prepped with NEBNext Ultra II DNA Library Prep Kit for Illumina (NEB, USA, E7645L) strictly following manufacturer's instructions, using a methylated adapter (E7536A). Without amplification post-library prep, the resulting library was purified and size-selected by 0.75X Ampure beads. Then, the eluted DNA was bisulfite-converted with EZ DNA Methylation-Lightning Kit (Zymo, USA, D5030T) and TrueMethyl oxBS-Seq Module without oxidation step (Nugen NUG_0414-32) following the manufacturer's instruction. The DNA obtained was amplified by PCR for 7-9 rounds with Q5U Hot Start High-Fidelity DNA Polymerase (NEB, M0515S) and purified by 0.8X Ampure beads. The samples were submitted to Macrogen (South Korea) for HiSeqX 150 bp paired-end sequencing according to standard Illumina cluster generation and sequencing protocols.

**Training datasets.** For training R9.4.1 models we used a high-quality ONT NA24385 dataset (https://labs.epi2me.io/gm24385-5mc/) and internally sequenced H1ESc PCR and M.SssI datasets. NA24385 dataset contains both ONT reads and Illumina reads produced using reduced representation bisulfite sequencing (RRBS). The dataset is a human B-lymphocyte cell line obtained from a white male. To produce a high-quality training dataset, we first aligned basecalled ONT reads on the human reference using minimap2. We kept only the reads with exactly one alignment with a mapping quality of 60 or more. Reads aligned to chromosomes 2-21, X and Y were used for training. Reads aligned to chromosome 22 were used for validation and reads aligned to chromosome 1 were used in the ablation study. The Bismark coverage

report used for extracting ground truth was generated by running the script provided in the dataset repository (see Section "Training datasets"). For base model training and validation, we chose positions with RRBS coverage of 50x or more. Moreover, we removed all positions that are partially methylated. ONT examples from chromosomes 2-21, X and Y that have been discarded due to the aforementioned filters have been stored for knowledge distillation training. At last, we randomly sampled 80 million training examples for training.

Because of a widely recognized bias in RRBS sequencing toward CpG-rich regions[73], we also sampled 10 million examples from the H1ESc PCR dataset and 10 million examples from the H1ESc M.SssI dataset. This resulted in a higher number of examples originating from CpG-poor regions. Both datasets were processed in the same manner as the previously mentioned NA24385 dataset. Instances originating from the PCR dataset are labeled as negatives, whereas instances from the M.SssI dataset are labeled as positives. This leads to a total of 100 million training examples. Furthermore, we included one million examples (500000 from each dataset) in the validation dataset.

For the R10.4.1 model training, we utilized the NA24385 data, which was the same sample used for the R9.4.1 training. High-quality ONT data was downloaded from the Oxford Nanopore Open Data project (Oxford Nanopore Technologies Benchmark Datasets was accessed on 2023-12-26 from https://registry.opendata.aws/ont-open-data). The ground truth was obtained using the same RRBS data as in the R9.4.1 training. The ONT data was basecalled using Dorado and aligned using minimap2. Following feature extraction, we associated the extracted examples with their respective labels by joining them on the reference position. We consider only positions that are either fully unmethylated or methylated, with a coverage of at least 30x. For the training dataset, we use chromosomes 1-21, X and Y, and chromosome 22 for validation. Additionally, we performed stratified sampling to ensure a balance of negative and positive data, yielding 188 million training examples and 1.9 million validation examples.

**Knowledge distillation datasets**. After R9.4.1 base model training, we performed knowledge distillation[74], a model compression method that results in similar or better performance compared to a teacher model while reducing the time needed for training and inference. The teacher model used for knowledge distillation is the trained Rockfish model trained on high-confident NA24385 data. To improve generalization and to introduce more biological diversity, we have sequenced two new mouse datasets (adult cardiomyocyte cell and adult blood cell) used during distillation training.

To build a dataset for knowledge distillation, we sampled 200 million examples: 100 million examples from the previously filtered data, and 50 million examples from each of the two mouse datasets. Next, we performed inference on the sampled data. Probabilities obtained from the teacher model were used as probabilities for knowledge distillation. Moreover, we added 50 million examples from the base model training (40 million from NA24385 and 10 million from synthetic H1ESc datasets). In total, we had 250 million examples of knowledge distillation training. All samplings used to build the knowledge distillation dataset were stratified samplings - there were exactly 125 million modified and 125 million unmodified examples.

**Evaluation datasets**. To show the robustness of our method, we performed an extensive evaluation of datasets corresponding to different human cell lines and organisms. To evaluate R9.4.1 models we used three B-lymphocyte cell lines: NA12878[19], NA19240[75], HX1[76], cancer cell line K562[34] and the newly sequenced human embryonic stem cell H1ESc. For R10.4.1, we used human NA12878 dataset (Oxford Nanopore Technologies Benchmark Datasets was accessed on 2024-01-02 from https://registry.opendata.aws/ont-open-data). We used all somatic chromosomes and chromosome X. For male samples (HX1, H1ESc) we also included chromosome Y. All reads were aligned to the

CHM13. Furthermore, we leveraged the newly sequenced C57BL/6 Neonatal dataset to assess the ability of ONT-based tools to generalize on non-human data. To evaluate performance on mouse data, we used all chromosomes present in the GRCm39 assembly. The first mouse replicate is used for evaluating R9.4.1 models, while for the R10.4.1 model evaluation, we merged four replicates (rep2 - rep5) and evaluated them together.

### Reporting summary
Further information on research design is available in the Nature Portfolio Reporting Summary linked to this article.

## Data availability
Both ONT and Illumina RRBS paired-end data for NA24385 is available via AWS at https://labs.epi2me.io/gm24385-5mc/. ONT NA12878 dataset is available via AWS at https://github.com/nanopore-wgs-consortium/NA12878. ONT data for NA12940 is available upon request from Chaisson et al.[75]. ONT data for HX1 is available at NCBI under project PRJNA533926. ONT data for K562 is available at Gene Expression Omnibus (GEO) under the BioProject GSE173688. WGBS paired-end data for NA12878 are available at ENCODE portal[77] under accession numbers ENCFF798RSS and ENCFF113KRQ (replicate 1), ENCFF585BXF and ENCFF851HAT (replicate 2). RRBS single-end data for NA12940 is available at ENCODE under accession numbers ENCFF000LZS (replicate 1) and ENCFF000LZT (replicate 2). WGBS paired-end data for HX1 is available at NCBI under the BioProject PRJNA301527. WGBS paired-end data for K562 is available at ENCODE under accession numbers ENCFF413KHN and ENCFF567DAI (replicate 1), ENCFF336KJH and ENCFF585HYM (replicate 2). All newly sequenced data (ONT and WGBS for H1ESc, ONT for mouse datasets, WGBS for neonatal mouse dataset) are available at NCBI under BioProject PRJNA876781. ChIP-seq data for NA12878 are available at ENCODE under accession numbers ENCSR000DZN (both two CTCF replicates and ChIP-seq control data). CHM13 excluded regions are available on GitHub (https://github.com/dozmorovlab/excluderanges). Gene annotations (http://courtyard.gi.ucsc.edu/~mhauknes/T2T/t2t_Y/annotation_set_v2/CHM13.v2.0.cat_liftoff_v2.gff3) and RepeatMasker annotations (https://t2t.gi.ucsc.edu/chm13/hub/t2t-chm13-v2.0/rmsk/rmsk.bigBed) are downloaded from UCSC Genome Institute (UCSC GI). GC content data and CpG island annotations are downloaded from UCSC Baskin School of Engineering (https://hgdownload.soe.ucsc.edu/hubs/GCA/009/914/755/GCA_009914755.4/bbi/GCA_009914755.4_T2T-CHM13v2.0.gc5Base.bw). Source data are provided with this paper.

## Code availability
Rockfish[78] code, including feature extraction, model layout, training and inference can be found at https://github.com/lbcb-sci/rockfish. All Rockfish models (R9.4.1 base, R9.4.1 small and R10.4.1) are available at https://zenodo.org/records/10867175. Moreover, they can be automatically downloaded using the script provided in the repository.

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

## Acknowledgements

This work has been supported in part by the Croatian Science Foundation under the project Single Genome and Metagenome Assembly (IP-2018-01-5886), by Epigenomics and Epitranscriptomics Research seed grant from the Genome Institute of Singapore (GIS), by the Career Development Fund (C210812037) from A*STAR. The computational work for this article was partially performed on resources of the National Supercomputing Centre, Singapore (https://www.nscc.sg) and A*STAR Computational Resource Centre. We would like to thank Dr. Matias Autio Ilmari, Dr. Matthew Andrew Ackers-Johnson, Ms. Wang Xiao Jenny, and Mr. Li Yiqing Peter for their help with the cell and tissue samples used in this paper and their helpful insight for this project. Moreover, thanks to Dr Yue Wan for her valuable comments on the manuscript and data presentation. Finally, we would also like to thank Mr. Low Hwee Meng and Ms. Leong See Ting from the GIS Integrated Genomics Platform core facility for their help and diligent work in Nanopore Sequencing.

## Author contributions

M.Š. and L.Z. conceived the project. D.S. designed and implemented Rockfish with help from M.Š. and L.Z.; L.Z. performed ONT and WGBS sequencing of human data and ONT sequencing of mouse data. D.S. and S.B. performed bioinformatics analysis and evaluation with the contribution of M.Š. and L.Z.; D.S., S.B., and M.Š. organized the manuscript. D.S., S.B., M.Š., and L.Z. wrote the manuscript with help from R.F.; M.Š. supervised the project. M.Š. and R.F. provided mentorship and support during the project.

## Competing interests

Mile Šikić leads the project AI-driven de novo diploid assembler jointly funded by AI Singapore and Oxford Nanopore Technologies. All other authors declare no competing interests.
