## [Peer Review File · Nature Communications]

Rockfish: A Transformer-based Model for Accurate 5-Methylcytosine Prediction from Nanopore SequencingReviewer #2 (Remarks to the Author):

As one of the original reviewers of a previous submission of this manuscript I appreciate many of the manuscript changes and response to the reviewers. I certainly see that this work contributes novel tools and to insight on how to use transformer-based architectures to effectively analyse nanopore data for the purpose of modified bascalling. However, the applicability of the work is severely hampered by the lack of analysis based on R10 flowcell data and models and thorough comparison with (ONT-based) models specific for the R10. I understand that the continuous updating of the pores and chemistries by ONT make it hard for the community to keep up to date and develop useful tooling. However, the R9 to R10 update happened quite some time ago and has been quite impactful. For instance, there has been recent but very substantial progress using the Dorado model, and even R10 remora is supposed to be better than R9 remora. The argument that there is no R10 training data is not true, as there is R10 GIAB data available (<https://labs.epi2me.io/giab-2023.05/>). I would really recommend the authors to include R10 results and benchmarks so as to increase utility for the community.

Reviewer #3 (Remarks to the Author): The reviewer co-reviewed with original Reviewer #1.

The manuscript introduces "Rockfish", a deep learning algorithm aimed at enhancing 5mC detection in DNA methylation studies via Nanopore sequencing. While the authors present a 5-percentage point improvement in single-base accuracy and F1 measure over other Nanopore-based methods, this enhancement appears relatively modest. The narrative could benefit from a more detailed exposition on the challenges Rockfish addresses, which would offer readers a clearer understanding of its advantages. The 'broad applicability' claim of Rockfish for studying 5mC methylation across various organisms and disease systems seems ambitious, especially in the absence of substantial evidence or specific examples. The manuscript would benefit from a more detailed and rigorous presentation of the algorithm's advantages and potential applications.

Moreover, the presented study, in its current iteration, does not seem to surpass the capabilities of contemporary tools. The proposed model's novelty, especially when compared to existing models like methBERT(10.1109/BIBM52615.2021.9669841), is not evident. The manuscript would benefit from addressing the following concerns:

Primary Concerns:

- (1) The deep learning architecture of the proposed approach bears a striking resemblance to methBERT (10.1109/BIBM52615.2021.9669841). The manuscript lacks a comprehensive comparison and discussion in terms of performance between the two, leaving questions about the novelty and superiority of Rockfish.
- (2) The manuscript does not specify which flowcells versions the proposed model supports. Given that advanced tools like Guppy and Dorado (<https://github.com/nanoporetech/dorado>) already cater to both 9.x and 10.x flowcells, it would be beneficial for the authors to highlight their model's unique features. Additionally, a performance comparison of Rockfish on 9.4 and 10.4 flowcells would be a valuable addition.
- (3) The manuscript omits a discussion on the support for Nanopore's latest POD5 format alongside the FAST5 format files, a feature present in tools like Guppy and DeepMod2 (<https://github.com/WGLab/DeepMod2>).
- (4) The data used for training is ambiguously described and not public available yet, making reproducibility a concern. Clearer details on the training data are essential.
- (5) The performance comparisons between the proposed method and other renowned methods are not directly comparable due to differences in training data. The manuscript could benefit from more evidence supporting cross-species validation.
- (6) A notable omission is the discussion on tools' ability (or lack thereof) to differentiate between 5hmC and 5mC, a significant advantage of nanopore sequencing over RRBS/WGBS. Tools like Remora, Guppy, and Dorado already possess this feature. An explanation or discussion on this aspect would be valuable.

Minor Concerns:

- (1) Line 52: The statement comparing the accuracy of methods using ONT raw signal with other methods should be more specific, especially when discussing basecalling and methylation calling for PacBio and Oxford Nanopore.
- (2) Line 77: The manuscript briefly touches upon DeepSignal2 without providing any reference.
- (3) Supplementary files: The inclusion of training scripts and training data sources for the proposed base and small model would be beneficial.

Rockfish - Response to the reviewers

We appreciate the feedback from the reviewers on our study. Below, we respond to each of their concerns in detail. Manuscript changes related to the comments are indicated with line numbers (e.g. L800). Manuscript text related to the changes is written in red.

Reviewer 2

Overview

As one of the original reviewers of a previous submission of this manuscript I appreciate many of the manuscript changes and response to the reviewers. I certainly see that this work contributes novel tools and to insight on how to use transformer-based architectures to effectively analyse nanopore data for the purpose of modified bascalling. However, the applicability of the work is severely hampered by the lack of analysis based on R10 flowcell data and models and thorough comparison with (ONT-based) models specific for the R10. I understand that the continuous updating of the pores and chemistries by ONT make it hard for the community to keep up to date and develop useful tooling. However, the R9 to R10 update happened quite some time ago and has been quite impactful. For instance, there has been recent but very substantial progress using the Dorado model, and even R10 remora is supposed to be better than R9 remora. The argument that there is no R10 training data is not true, as there is R10 GIAB data available (<https://labs.epi2me.io/giab-2023.05/>). I would really recommend the authors to include R10 results and benchmarks so as to increase utility for the community.

Response

Thank you for your feedback and suggestions. We agree that including R10 models (and comparing them with Remora) would benefit the community. Additionally, we're grateful for the link to the R10 data you provided. We've opted to train Rockfish on R10.4.1 5kHz data and to assess its performance against Remora using two distinct datasets—one human and one newly sequenced mouse dataset. Furthermore, we've also added support for POD5 files, the current ONT raw data format. (L18-21, L78, L94-95, L122-131, L139, L149-152, Fig 1a, L153-156, Fig 2a, L165-169, L178-179, L181-182, L189-190, L198-199, L203-205, L207-208, Fig 3a, Fig 3d-e, L228-229, L235-236, L251-256, L264-268, Fig 4, L269-324, Fig 5a, L335-340, L398-399, L410, L415-418, L428-429, L435-438, L446-463, L482, L628-L632, L642-645, L650-651, L671-673, L831-835, L842-843, L944-953, L973-974, L978-980; Supplementary Tables 1-3, Supplementary Tables 6-11, Supplementary Figure S1, Supplementary Figures S4-S5, Supplementary Figure S8, Supplementary Figure S10, Supplementary Material)

Reviewer 3

Overview

The manuscript introduces "Rockfish", a deep learning algorithm aimed at enhancing 5mC detection in DNA methylation studies via Nanopore sequencing. While the authors present a 5-percentage point improvement in single-base accuracy and F1 measure over other Nanopore-based methods, this enhancement appears relatively modest. The narrative could benefit from a more detailed exposition on the challenges Rockfish addresses, which would offer readers a clearer understanding of its advantages. The 'broad applicability' claim of Rockfish for studying 5mC methylation across various organisms and disease systems seems ambitious, especially in the absence of substantial evidence or specific examples. The manuscript would benefit from a more detailed and rigorous presentation of the algorithm's advantages and potential applications.

Moreover, the presented study, in its current iteration, does not seem to surpass the capabilities of contemporary tools. The proposed model's novelty, especially when compared to existing models like methBERT(10.1109/BIBM52615.2021.9669841), is not evident. The manuscript would benefit from addressing the following concerns:

Response

We sincerely appreciate the time you've taken to provide us with your feedback. Below, we've addressed each of your comments in detail, outlining our responses and any corresponding actions we plan to take.

Reviewer Point 1 P3.1

The deep learning architecture of the proposed approach bears a striking resemblance to methBERT (10.1109/BIBM52615.2021.9669841). The manuscript lacks a comprehensive comparison and discussion in terms of performance between the two, leaving questions about the novelty and superiority of Rockfish.

Response

We value your feedback regarding the comparison with methBERT. While our model does share certain architectural elements with methBERT (Transformer encoder), we respectfully disagree with the assertion that our method "bears a striking resemblance to methBERT". Here are our key points of differentiation:

1. MethBERT uses Tombo re-squigglng for signal-to-sequence alignment, leading to a notable slowdown. In contrast, Rockfish relies on deep learning, specifically multi-head attention in Transformer, for this alignment process. (Reference from manuscript to be added)
2. Following re-squigglng, methBERT extracts "event-based" features, akin to deepMOD, which primarily involve statistical metrics such as mean, standard deviation, length, and nucleotide information. We contend that relying solely on statistics may lead to information loss. Instead, Rockfish adopts a different approach by directly feeding raw signals into the deep-learning model. This strategy aims to minimize inductive bias and enable the learning of potentially valuable transformations, including statistical features such as signal mean. (Reference from manuscript to be added)

- While methBERT uses only a Transformer encoder to derive contextual representations of each event, our approach utilizes both an encoder and a decoder. This enables us to first establish a contextual representation of each signal block (encoder) and then align these blocks with the sequence to extract modification information (decoder). (Reference from manuscript to be added)

To summarize, our approach differs from methBERT in several key aspects, including the feature extraction pipeline, the input features, and the deep-learning methodology employed for modification prediction.

We decided against evaluating methBERT due to:

- The model was trained and evaluated only on the (synthetic) R9 data. The R9 pore model is deprecated in favour of the R9.4.1 model.
- While the authors do not give any comparison with other tools in the published article a comparison was made between methBERT and Megalodon in the pre-print, with the conclusion that Megalodon outperforms methBERT. Since we evaluated Megalodon within our manuscript, we see no necessity to independently evaluate methBERT.

To validate the accuracy of our claims, we conducted an evaluation of methBERT on the NA12878 dataset. Given the computational demands of Tombo, our initial assessment was limited to this dataset. This allowed us to ensure the correctness of the statements made previously.

Read-level results:

Tool	Count	Accuracy	Precision	Recall	FPR	F1
Rockfish	240121846	0.937895	0.938064	0.939494	0.063749	0.938778
methBERT	240121846	0.872828	0.892287	0.851924	0.105689	0.871638

Site-level results:

Tool	Count	Accuracy	Precision	Recall	FPR	F1
Rockfish	15077584	0.993948	0.991256	0.996715	0.008832	0.993978
methBERT	15077584	0.980512	0.987078	0.973858	0.012805	0.980424

Pearson's correlation: 0.863323 for methBERT and 0.899044 for Rockfish.

Taking into consideration all previously discussed points and results on NA12878, we deem that further methBERT evaluation is not needed.

Reviewer Point 2 P3.2

The manuscript does not specify which flowcells versions the proposed model supports. Given that advanced tools like Guppy and Dorado (<https://github.com/nanoporetech/dorado>) already cater to both 9.x and 10.x flowcells, it would be beneficial for the authors to highlight their model's unique features. Additionally, a performance comparison of Rockfish on 9.4 and 10.4 flowcells would be a valuable addition.

Response

Thank you for your feedback on the necessity of specifying the chemistry type used in the evaluation and the importance of evaluating Rockfish on R10.4.1. We acknowledge the value of including the R10.4.1 evaluation in the manuscript. Consequently, we have conducted evaluations of Rockfish using two R10.4.1 datasets (human and mouse) and compared its performance against that of Remora. Additionally, we have taken care to differentiate between R9.4.1 and R10.4.1 datasets and included any other information specific to the chemistry type. (L18-21, L78, L94-95, L122-131, L139, L149-152, Fig 1a, L153-156, Fig 2a, L165-169, L178-179, L181-182, L189-190, L198-199, L203-205, L207-208, Fig 3a, Fig 3d-e, L228-229, L235-236, L251-256, L264-268, Fig 4, L269-324, Fig 5a, L335-340, L398-399, L410, L415-418, L428-429, L435-438, L446-463, L482, L628-L632, L642-645, L650-651, L671-673, L831-835, L842-843, L944-953, L973-974, L978-980; Supplementary Tables 1-3, Supplementary Tables 6-11, Supplementary Figure S1, Supplementary Figures S4-S5, Supplementary Figure S8, Supplementary Figure S10, Supplementary Material)

Reviewer Point 3 P3.3

The manuscript omits a discussion on the support for Nanopore's latest POD5 format alongside the FAST5 format files, a feature present in tools like Guppy and DeepMod2 (<https://github.com/WGLab/DeepMod2>).

Response

We appreciate your input regarding the discussion on the POD5 file format. We have considered your feedback and have decided to incorporate support for the POD5 format. (L630, Supplementary Material)

Reviewer Point 4 P3.4

The data used for training is ambiguously described and not public available yet, making reproducibility a concern. Clearer details on the training data are essential.

Response

Thank you for your comment regarding the training data description. For the newly sequenced datasets, there is the description of:

1. Preparation and sequencing (L628-632, L856-920)
2. Basecalling (Guppy 5.0.14 for all R9.4.1 datasets, Dorado 0.4.2 for R10.4.1 datasets; L842-843, Supplementary Material)
3. Feature extraction and training/validation size and class distribution (L922-968)

We recognize the necessity for more detailed training steps. Therefore, we have included an example of how to generate a training (or validation) dataset in the Supplementary Material.

Since one of the project's contributions is the publication of newly sequenced datasets, a common practice is to make them available upon publication. However, to make the process smoother, we have decided to publicly release all the data. The BioProject ID is PRJNA876781.

Reviewer Point 5 P3.5

The performance comparisons between the proposed method and other renowned methods are not directly comparable due to differences in training data. The manuscript could benefit from more evidence supporting cross-species validation.

Response

We appreciate your feedback regarding the comparison with other methods and cross-species validation. We have conducted thorough testing of ONT-based tools on six R9.4.1 datasets and two R10.4.1 datasets, exceeding the number tested in benchmark papers. Altogether, these eight datasets include:

1. Different chemistry types – R9.4.1 and R10.4.1
2. Different organisms – human and mouse
3. Different cell lines - B-lymphocyte, cancer cell line, embryonic stem cell (human); cardiomyocyte (mouse)

We deem that these eight datasets exhibit sufficient variation to validate the models' generalization capabilities. Additionally, certain evaluations were conducted on newly sequenced data to ensure fairness in comparison. Re-training all models using identical data is impractical due to computational constraints (such as hyperparameter optimization), logistical challenges (such as the absence of publicly available training scripts) and bias in dataset selection (e.g. model's complexity dictates the size of training data). Due to all the aforementioned issues, we believe that this task is beyond the scope of this project.

Reviewer Point 6 P3.6

A notable omission is the discussion on tools' ability (or lack thereof) to differentiate between 5hmC and 5mC, a significant advantage of nanopore sequencing over RRBS/WGBS. Tools like Remora, Guppy, and Dorado already possess this feature. An explanation or discussion on this aspect would be valuable.

Response

Thank you for your comment. We acknowledge that this is an important discussion point, and we've opted to briefly address it in the discussion section. (L484-490)

Reviewer Point 7 P3.7 (Minor comments)

1. Line 52: The statement comparing the accuracy of methods using ONT raw signal with other methods should be more specific, especially when discussing basecalling and methylation calling for PacBio and Oxford Nanopore.
2. Line 77: The manuscript briefly touches upon DeepSignal2 without providing any reference.
3. Supplementary files: The inclusion of training scripts and training data sources for the proposed base and small model would be beneficial.

Response

1. We have included references to two related works that describe the advantages and pitfalls of different sequencing technologies for the detection of DNA methylations. Ni et al. (2022) compare the results of predicting 5mC from PacBio CSS reads to predictions

from WGBS and Nanopore, while Kong et al. (2023) analyse the potential sources of false positive and false negative calls depending on the sequencing technologies. Both works show that methylation calling from PacBio CSS exhibits more technology-intrinsic biases that result in less accurate methylation prediction. We do not expand the explanation in the main text to avoid an unnecessarily lengthy introduction.

2. We have included a reference to the DeepSignal2 GitHub repository (L81).
3. We've included an example outlining the steps involved in model training and have also included a sample script for each step (Supplementary Material).

Reviewer #2 (Remarks to the Author):

I would like to thank the authors for the careful consideration of the reviewer remarks. The step to R10 and compatibility with POD5 substantially improve the utility of Rockfish. The response regarding the comparison with methBERT is acceptable. I have no further comments.

Reviewer #2 (Remarks on code availability):

There is ample instruction for installing and using the code. All ok for me.

Reviewer #3 (Remarks to the Author):

The authors have made an effort to incorporate feedback on their manuscript, 'Rockfish: A Transformer-based Model for Accurate 5-Methylcytosine Prediction from Nanopore Sequencing'. However, upon review of the revised manuscript, it appears that not all of the previously raised concerns have been adequately addressed, highlighting certain limitations as follows:

(1) It appears that further exploration into the detection of 5-hydroxymethylcytosine (5hmC) could enhance the study. Given that 5hmC detection capabilities, such as those offered by ONT Megalodon and Guppy, mark a significant advancement of Nanopore sequencing over traditional RRBS, incorporating a study on 5hmC could potentially underscore the novelty and applicability of your model.

(2) The manuscript would greatly benefit from a comparison between the Rockfish model and the Guppy 5mC and 5hmC models, specifically versions R9.4.1 and R10.4.1, which are noted for their high performance. Such a comparison is crucial for establishing the relative performance of Rockfish and for facilitating replication by other researchers. Additionally, clarity regarding the specific models used for Megalodon Rerio and Remora, with reference to the provided GitHub link, is necessary for ensuring the reproducibility of your results.

(3) The inclusion of results comparing Rockfish to similar models, such as methBert and DeepMod2, directly within the main figures would be advantageous. Considering the advancements detailed in recent publications (i.e., <https://www.nature.com/articles/s41467-024-45778-y>), a direct comparison could highlight Rockfish's unique features and advantages. Moreover, addressing the robustness of DeepMod2, which has been validated across different species, could provide valuable insights into the potential applicability and versatility of Rockfish.

(4) Overall, the performance of Rockfish seems not too much improvement. Highlighting any areas where Rockfish excels or discussing potential refinements to enhance its performance could provide a clearer understanding of its contribution to the field of 5-Methylcytosine prediction from Nanopore sequencing.

I recommend that the authors fully address the aforementioned comments to enhance the manuscript's suitability for publication.

Rockfish – Response to the Reviewers

We appreciate the feedback from the reviewers on our study. Below, we respond to each of their concerns in detail. Manuscript changes related to the comments are indicated with line numbers (e.g. L800). Manuscript text related to the changes is written in red.

Reviewer 2

Overview

I would like to thank the authors for the careful consideration of the reviewer remarks. The step to R10 and compatibility with POD5 substantially improve the utility of Rockfish. The response regarding the comparison with methBERT is acceptable. I have no further comments.

Response

We are sincerely grateful for your comments and suggestions during this review process.

Reviewer 3

Overview

The authors have made an effort to incorporate feedback on their manuscript, ‘Rockfish: A Transformer-based Model for Accurate 5-Methylcytosine Prediction from Nanopore Sequencing’. However, upon review of the revised manuscript, it appears that not all of the previously raised concerns have been adequately addressed, highlighting certain limitations as follows:

Response

We appreciate your feedback. Below, we've addressed each of your comments in detail, outlining our responses and any corresponding actions we plan to take.

Reviewer Point 1 P3.1

It appears that further exploration into the detection of 5-hydroxymethylcytosine (5hmC) could enhance the study. Given that 5hmC detection capabilities, such as those offered by ONT Megalodon and Guppy, mark a significant advancement of Nanopore sequencing over traditional RRBS, incorporating a study on 5hmC could potentially underscore the novelty and applicability of your model.

Response

We appreciate the reviewer's suggestion to include a study on 5hmC to strengthen the novelty and applicability of our model. The primary focus of this work was to develop a new, state-of-the-art method for detecting 5mC modifications from the Nanopore data.

We argue that the contributions presented in this work are sufficient even without a 5hmC study. Nevertheless, we recognize the significance of 5hmC in advancing the capabilities of Nanopore sequencing and would perform a 5hmC study if it was feasible - the reason for not including the 5hmC study is the lack of high-quality (ONT + BS + oxBS) data. This was discussed in the last version of the manuscript (L499-500) and was highlighted in the response to the reviewers' comments. Moreover, a recently published method, DeepMod2, states the same reason for not providing a 5hmC model (Page 15).

Reviewer Point 2 P3.2

The manuscript would greatly benefit from a comparison between the Rockfish model and the Guppy 5mC and 5hmC models, specifically versions R9.4.1 and R10.4.1, which are noted for their high performance. Such a comparison is crucial for establishing the relative performance of Rockfish and for facilitating replication by other researchers. Additionally, clarity regarding the specific models used for Megalodon Rerio and Remora, with reference to the provided GitHub link, is necessary for ensuring the reproducibility of your results.

Response

We value the reviewer's comments regarding the Guppy evaluation and the suggestion to add information about the methods and models used in the evaluation.

While we recognize the relevance of comparing our R9.4.1 models against Guppy, we decided to focus on comparisons with the "Megalodon Rerio" and "Megalodon Remora" pipelines for several reasons. "Megalodon Rerio" pipeline consists of Rerio research model used for joint canonical and modification basecalling and Megalodon for further refinement of modification calls. In multiple benchmark studies, it was shown that the "Megalodon Rerio" pipeline outperforms Guppy models (<https://www.nature.com/articles/s41467-021-23778-6> and <https://genomebiology.biomedcentral.com/articles/10.1186/s13059-021-02510-z>).

"Megalodon Remora" pipeline consists of Guppy for canonical basecalling, Remora for modification calling and Megalodon for refinement. In 2022 ONT announced Remora, a deep-learning model that performs a second pass for modification calling, moving away from the joint canonical and modification basecalling (done by Guppy). Moreover, Guppy basecaller is deprecated in favour of Dorado. In Dorado, Remora models are used to perform separate modification calling after canonical basecalling. Dorado framework (with the default Remora model for v4.2.0) is used for evaluating the R10.4.1 model. Given these developments, we deem that a direct comparison to older Guppy models is less relevant to the objectives of our study.

We have included a discussion in the manuscript to explain these choices and provide context on the technological evolution in nanopore sequencing. This discussion can be found in the manuscript (L94-99).

Regarding the versions and references for the models used (Megalodon Remora and Megalodon Rerio), we have provided the following information in our manuscript:

1. Megalodon - version (v2.4.2; L861-862, Supplementary Material). We acknowledge the omission of a reference to the GitHub repository.
2. Rerio - GitHub link as a reference (Reference 30); version of the model used (Footnote 6 on page 23, Supplementary Material)
3. Remora - GitHub link as a reference (Reference 29); version of the tool used (v0.1.2; L862, Supplementary Material with an example command for running “Megalodon Remora” pipeline).

To enhance the reproducibility of our results, we have now explicitly added the following information:

1. Reference to Dorado GitHub repository (Reference 28)
2. Reference to Megalodon GitHub repository (Reference 33)
3. R9.4.1 Remora model version added as a Footnote 5 on page 23 (included in the Megalodon example command in the Supplementary Information)
4. R10.4.1 Remora model version added as a Footnote 7 on page 23 (included in the Dorado example command in the Supplementary Information).

Reviewer Point 3 P3.3

The inclusion of results comparing Rockfish to similar models, such as methBert and DeepMod2, directly within the main figures would be advantageous. Considering the advancements detailed in recent publications (i.e., <https://www.nature.com/articles/s41467-024-45778-y>), a direct comparison could highlight Rockfish’s unique features and advantages. Moreover, addressing the robustness of DeepMod2, which has been validated across different species, could provide valuable insights into the potential applicability and versatility of Rockfish.

Response

We acknowledge your suggestion to include direct comparisons with models like methBert and DeepMod2 in our main figures to underscore Rockfish’s unique features.

We did an initial comparison of Rockfish with methBERT on the NA12878 dataset and showed that methBERT underperforms on both read- and site-level when compared with Rockfish and other tools evaluated in our study. Since methBERT incorporates Tombo re-squiggling into its pipeline, due to the time and memory demands Tombo imposes, we decided against further evaluations of methBERT. For the same reasons, we decided not to evaluate any tools requiring Tombo as a pre-processing step, including previously mentioned methBERT and DeepSignal (L116-119). Comparison with tools depending on Tombo preprocessing was also excluded from recent DeepMod2 publication for the same reasons (Page 3).

Since DeepMod2 was published during our revision, it was not included in the evaluation. Notably, the R9.4.1 Rockfish model was evaluated in the DeepMod2 paper and found to perform on par with or better than DeepMod2 and other competitive tools: “For R9.4.1 datasets of HG002, HG003 and HG004, DeepMod2 BiLSTM and Rockfish consistently outperform other models, with genome-wide F1-score in a narrow range of 99.85–99.92%. All tools perform slightly better on human genomes than on the mouse genome NIH3T3, with Rockfish performing best at 99% F1-score, followed by

DeepMod2 at 98.75%. ...“. It is also worth noticing that the training of DeepMod2 was done using samples from chr2-21 of three datasets, HG002, HG003, and HG004, corresponding to the same cell type, while the evaluation was performed on chr1 of the same datasets and a mouse dataset for the R9.4.1 pore version. On the contrary, Rockfish was trained using only NA24385 (HG002) dataset and evaluated on 6 different R9.4.1 and 2 different R10.4.1 datasets which included multiple cell types and species that were not seen during the training. Rockfish outperforms competitive tools across those datasets showing its robustness and generalization potential. This claim is also supported by evaluation results presented in DeepMod2 where Rockfish outperforms other tools across human and mouse evaluation datasets. We have addressed these points in the revised version of the manuscript (L119-126).

We emphasize the key points of differentiation between Rockfish and existing models that incorporate Transformer-based architectures in their pipeline (i.e., methBERT and DeepMod2) once again. In addition to methBERT’s dependency upon deprecated Tombo re-squiggling:

- methBERT and DeepMod2 rely on the extraction of “event-based” features, primarily involving statistical descriptors such as mean, standard deviation, length, and basecalled nucleotide information. Instead, Rockfish adopts a different approach by directly feeding raw signals into the deep-learning model. This strategy aims to minimize inductive bias and enable the learning of potentially valuable transformations, including statistical features such as signal mean.
- methBERT and DeepMod2 only utilize a Transformer encoder to derive contextualized representations of events while Rockfish utilizes both an encoder and a decoder. This allows contextualized representations of signal blocks obtained through the encoder to be aligned with the sequence to extract modification information in the decoder.

Reviewer Point 4 P3.4

Overall, the performance of Rockfish seems not too much improvement. Highlighting any areas where Rockfish excels or discussing potential refinements to enhance its performance could provide a clearer understanding of its contribution to the field of 5-Methylcytosine prediction from Nanopore sequencing.

Response

Thorough evaluation of Rockfish across different cell types, species and genomic contexts showed that Rockfish outperformed existing tools on R9.4.1 and R10.4.1 datasets. Rockfish showed improvement in accuracy and F1 score of up to 5 percentage points for R9.4.1 datasets, resistance to class imbalance and consistency across different cell types and species. When evaluated against the state-of-the-art Remora on R10.4.1 datasets, Rockfish showed improvement in detecting 5mC in a human and a mouse dataset. Furthermore, our analysis of the effects that read-level predictions have on site-level results shows that not only does Rockfish exhibit higher average coverage compared with other ONT-based tools, but it also maintains high performance in important genomic regions such as promoters. Analysis of Remora’s filtering of

uncertain calls unveils bias towards filtering uncertain predictions in CpG-rich regions, including CpG-rich promoters where Remora achieves notably worse results compared with Rockfish. The evaluations presented in the recent DeepMod2 publication support the strengths of Rockfish when evaluated on different species and in different contexts. Rockfish serves as a suitable baseline for further research ventures aimed at developing deep-learning (especially attention-based) methods for detecting 5mC (and other types of modifications) from Nanopore signal. Lastly, given that Rockfish is an open-source tool, other members of the scientific community can easily implement potential changes and improvements. We deem all these results as a valuable contribution to the field of 5-methylcytosine prediction from Nanopore sequencing.